# MTL-KD: Multi-Task Learning Via Knowledge Distillation for Generalizable Neural Vehicle Routing Solver

**Yuepeng Zheng**[1][*], **Fu Luo**[2,3][*], **Zhenkun Wang**[2,3], **Yaoxin Wu**[4], **Yu Zhou**[1][†]

[1] College of Computer Science and Software Engineering,
Shenzhen University, Shenzhen, China
[2] School of Automation and Intelligent Manufacturing,
Southern University of Science and Technology, Shenzhen, China
[3] Guangdong Provincial Key Laboratory of Fully Actuated System Control Theory and Technology,
Southern University of Science and Technology, Shenzhen, China
[4] Department of Industrial Engineering and Innovation Sciences,
Eindhoven University of Technology, Eindhoven, The Netherlands
`2019271024@email.szu.edu.cn,luof2023@mail.sustech.edu.cn`
`wangzhenkun90@gmail.com,y.wu2@tue.nl, zhouyu_1022@126.com`

## Abstract

Multi-Task Learning (MTL) in Neural Combinatorial Optimization (NCO) is a promising approach for training a unified model capable of solving multiple Vehicle Routing Problem (VRP) variants. However, existing Reinforcement Learning (RL)-based multi-task methods can only train light decoder models on small-scale problems, exhibiting limited generalization ability when solving large-scale problems. To overcome this limitation, this work introduces a novel multi-task learning method driven by knowledge distillation (MTL-KD), which enables efficient training of heavy decoder models with strong generalization ability. The proposed MTL-KD method transfers policy knowledge from multiple distinct RL-based single-task models to a single heavy decoder model, facilitating label-free training and effectively improving the model's generalization ability across diverse tasks. In addition, we introduce a flexible inference strategy termed Random Reordering Re-Construction (R3C), which is specifically adapted for diverse VRP tasks and further boosts the performance of the multi-task model. Experimental results on 6 seen and 10 unseen VRP variants with up to 1,000 nodes indicate that our proposed method consistently achieves superior performance on both uniform and real-world benchmarks, demonstrating robust generalization abilities. The code is available at https://github.com/CIAM-Group/MTLKD.

## 1 Introduction

The Vehicle Routing Problem (VRP) is a classical combinatorial optimization problem (COP) with widespread application across numerous real-world scenarios, including logistics distribution, traffic scheduling, and emergency response [1–5]. Obtaining exact VRP solutions is computationally difficult due to its NP-hard nature [6]. Traditional heuristic algorithms [7–9] exhibit superior performance on many classic VRP tasks. However, they typically necessitate substantial domain-specific knowledge for design. In recent years, neural combinatorial optimization (NCO) [10–20] has emerged as a

---

[*]Equal contributors
[†]Corresponding author

promising approach for tackling VRPs. This approach can automatically learn problem-solving strategies using neural networks and has the potential to minimize reliance on specialized domain knowledge. This field has developed rapidly, with some NCO algorithms approaching or even surpassing the performance of classical heuristic algorithms on some COPs [21, 22]. However, when solving different tasks, many NCO methods require modifying the model components for each individual task and then retraining the model, which significantly restricts their versatility in handling diverse VRP variants.

Recently, considerable research efforts have been devoted to developing unified models capable of solving multiple VRP variants using Multi-Task Learning (MTL) [23–25]. The majority of unified models for VRP variants adopt a heavy encoder-light decoder (HELD) architecture. These models have relatively low computational costs and can be directly trained using RL, demonstrating excellent performance on small-scale problems. However, their generalization ability significantly deteriorates when solving with large-scale instances, primarily because the light decoder struggles to extract sufficient information from the high-density and complex node embeddings [26].

On the other hand, the light encoder-heavy decoder [27–29] architecture demonstrates excellent generalization capabilities on large-scale problems. Its heavy decoder contributes to its strong scale generalization performance, which re-evaluates the relationships between the remaining nodes and first and last nodes during the iterative decoding process. Leveraging the heavy decoder architecture has the potential to achieve good scale generalization using a multi-task unified model. However, the substantial memory and computational demands inherent in heavy decoders render the model training using Reinforcement Learning (RL) impractical. Employing Supervised Learning (SL) is also difficult due to the absence of labeled data for multiple VRP variants. While SIT [29] employs a self-improved Training method using operations like local reconstruction [27] to refine model-generated solutions as training labels, it suffers from the generation of numerous low-quality labels during early and mid-training, leading to a protracted process and an increased training burden. We provide a comprehensive literature review on multi-task neural solvers for VRPs in Appendix A.

To address the limitation associated with the training for the heavy decoder-based model on multiple VRP variants, we introduce a novel multi-task learning method driven by knowledge distillation (MTL-KD). The proposed MTL-KD method transfers policy knowledge from multiple distinct RL-based single-task models to a single heavy decoder model, facilitating label-free training and effectively improving the model's generalization ability across diverse tasks. Furthermore, during inference, we propose a general Random Reordering Reconstruction strategy for various variants of the VRP. By randomly reordering the external order of subtours, R3C significantly enhances solution sampling diversity, mitigates the risk of getting trapped in local optima, and further improves performance. Our contributions are summarized as follows: 1) We achieve efficient label-free training of heavy decoder models for multi-task VRPs through the proposed MTL-KD method. 2) We propose a novel R3C strategy to further enhance the performance of the multi-task model. 3) The MTL-KD model demonstrates excellent performance on 6 seen training tasks and 10 unseen tasks with up to 1,000 nodes, and real-world datasets, exhibiting good scale generalization ability and significantly outperforming existing multi-task VRP models.

## 2 Preliminaries

**Capacitated Vehicle Routing Problem and Its Variants**   The Capacitated Vehicle Routing Problem (CVRP) is typically defined as follows: Given a central depot $v_0$ and $N$ customer nodes $v_1$ to $v_n$, all interconnected with distances $e_{ij}$. Each customer $i$ has a demand $d_i$, and all vehicles depart from the depot, serve customers, and return to the depot. Each vehicle has a capacity $C$, satisfying $C > d_i$. All customers must be visited exactly once, and the total load of the vehicle during its route must not exceed its capacity. The objective of CVRP is to minimize the total travel distance of all vehicles.

By introducing additional constraints, CVRP can be extended into various variants. The four types of additional constraints studied in this paper include: **(1) Open Route (O):** Vehicles are not required to return to the depot after completing their service. **(2) Time Windows (TW):** Each node $i$ has a service time window $[s_i, l_i]$, within which vehicles must arrive at the customer (early arrivals require waiting until the earliest service start time). **(3) Backhaul (B):** Customer demands can be positive (linehauls) or negative (backhauls), and there is no restriction on the sequence of visiting linehauls and backhauls customers. **(4) Duration Limit (L):** The total travel distance of each vehicle must not

exceed a predefined threshold $L_{thre}$. Combining these constraints with CVRP can form 16 different VRP variants, with specific details provided in Appendix B.

**Solution Construction Process by Neural Solver** Constructive neural solvers [10, 11, 27] typically employ an encoder-decoder architecture and construct solutions in an autoregressive manner. For a given VRP instance $\mathcal{G} = (V, E)$, where $V$ denotes the set of nodes, including node features such as demand, and $E$ represents the set of edges, often formed by a distance matrix. The encoder is used to extract node information and generate embedding vectors for each node, while the decoder constructs the next node based on the current partial solution state $s_{t-1}$. The decoder is subject to node constraints during the construction process to avoid generating infeasible solutions. The entire solving process can be represented as:

$$\pi_{\boldsymbol{\theta}}(\tau|\mathcal{G}) = \prod_{t=1}^{\ell} \pi_{\boldsymbol{\theta}}(a_t|s_t, \mathcal{G}),$$

where $\tau$ is the complete solution, $a_t$ is the action (node) selected at time step $t$, $\ell$ is the total number of actions, and $\pi_{\boldsymbol{\theta}}$ represents the neural solver parameterized by $\boldsymbol{\theta}$. The decoder continuously constructs nodes until all nodes have been visited, forming a complete solution. Through training, the model learns how to effectively select the next node based on the current partial solution, thereby optimizing the quality of the entire solution.

**Knowledge distillation** Knowledge Distillation [30] is a model compression and knowledge transfer technique aimed at transferring the knowledge learned by a high-performing, pre-trained teacher model to a smaller student model. The core idea is to achieve this transfer by minimizing the discrepancy between the outputs of the teacher and student models. The student model's training loss typically includes a distillation loss ($\mathcal{L}_{KD}$) that measures the difference between the student's predictions and the teacher's soft targets, generally the KL divergence $\mathcal{L}_{KD} = \text{KL}(\pi_T || \pi_S)$. Additionally, depending on the specific task requirements, an original task loss ($\mathcal{L}_{Task}$) can optionally be included, and the total loss ($\mathcal{L}$) for the student model can be expressed as:

$$\mathcal{L} = \alpha \mathcal{L}_{KD} + (1 - \alpha) \mathcal{L}_{Task},$$

where $\alpha$ is a weight parameter. In the field of NCO, knowledge distillation has been used to achieve model lightweighting and can improve the generalization ability of models on unseen problems to a certain extent [31–34].

## 3 Multi-Task Learning Via Knowledge Distillation

In this section, we propose a multi-task learning framework via knowledge distillation to address the challenge of applying heavy decoder models in the multi-task domain due to their reliance on large amounts of labeled data. The proposed framework effectively enhances the model's generalization capabilities across different tasks and scales.

### 3.1 Framework Overview

Our proposed multi-task training framework via knowledge distillation aims to enhance the model's generalization capabilities across different tasks and scales. As depicted in Figure 1, the framework first categorizes VRP variants into seen and unseen tasks for model training. Given $N$ seen tasks, we first pre-train $N$ individual teacher models employing a Heavy Encoder-Light Decoder architecture [11]. Subsequently, we construct a multi-task heavy decoder student model and train it using knowledge distillation, leveraging the output distributions of the teacher models as supervision signals.

### 3.2 Teacher Model Training

For each seen VRP variant task, we first generate the corresponding instance data. Then, we independently train a teacher model for each task, which adopts the POMO [11] structure and serves as a policy network. We optimize the parameters of the teacher models using the policy

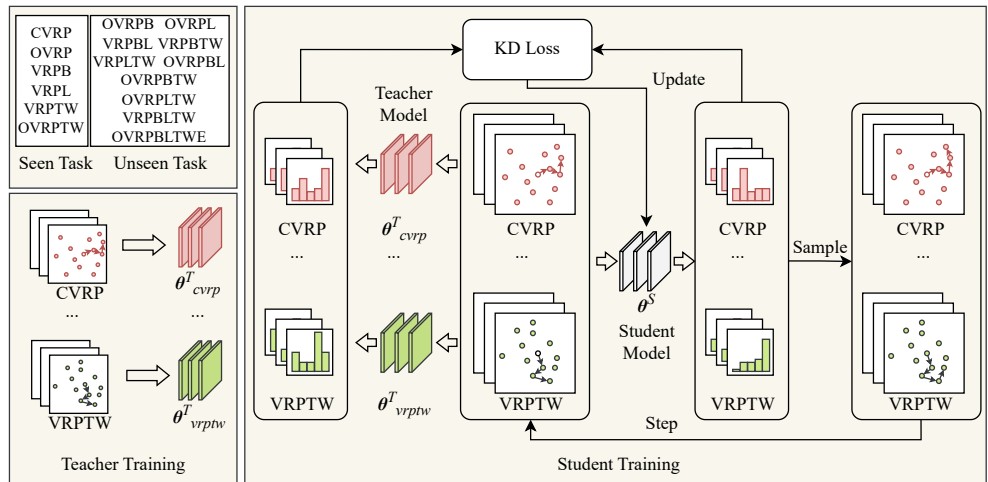

Figure 1: Framework for Multi-Task Training via Knowledge Distillation. Top-left: seen and unseen tasks. Bottom-left: Independent training of teacher models. Right: Student model distillation via teacher output distributions.

gradient method in RL, with the objective of maximizing the reward (negative tour length). The loss function [35] can be expressed as:

$$\mathcal{L}(\boldsymbol{\theta}^T) = -\mathbb{E}_{\tau \sim \pi_{\boldsymbol{\theta}^T}(\cdot|\mathcal{G})}[(r(\tau) - b^T(\mathcal{G})) \log \pi_{\boldsymbol{\theta}^T}(\tau|\mathcal{G})],$$

where $b^T(\mathcal{G})$ is the baseline (average reward of multi-start trajectories for the instance), $\pi_{\boldsymbol{\theta}^T}$ is the teacher model's policy distribution, $\boldsymbol{\theta}^T$ represents the parameters of the teacher model, and $r$ is the reward.

### 3.3 Student Model Training

During the training phase of the student model, we employ a multi-task heavy decoder architecture that processes instances from all $M$ seen problem types simultaneously. For each training batch, we generate instances from all $M$ seen problem types and process them together. At each decoding step $t$, the student model parameterized by $\boldsymbol{\theta}^S$ outputs a probability distribution $\pi_{\boldsymbol{\theta}^S}(a_t|s_t, \mathcal{G})$ over the next node to be selected.

For knowledge distillation, we feed each problem instance to its corresponding pre-trained teacher model parameterized by $\boldsymbol{\theta}^{T_m}$, $m \in \{1, \ldots, M\}$ to obtain the teacher's output distribution $\pi_{\boldsymbol{\theta}^{T_m}}(a_t|s_t, \mathcal{G})$ at the same decoding step, ensuring that each problem instance is supervised by its corresponding specialized teacher model.

The knowledge distillation loss at training step $t$ is computed as the Kullback-Leibler divergence between the student model's output distribution and the corresponding teacher model's output distribution:

$$\mathcal{L}_{KD}^{(t)} = \sum_{m=1}^{M} \text{KL}(\pi_{\boldsymbol{\theta}^{T_m}}(a_t|s_t, \mathcal{G}) \| \pi_{\boldsymbol{\theta}^S}(a_t|s_t, \mathcal{G})).$$

By minimizing this loss function, the student model learns to mimic the behavior of multiple teacher models simultaneously, thereby acquiring generalized knowledge from different problem types. After each decoding step, the algorithm selects the next node based on the student model's probability distribution, transitions to the new state, and repeats this knowledge distillation process.

# 4 Architecture of Generalizable Multi-Task Neural Solver

Heavy decoder models have demonstrated remarkable scale generalization performance on single-task VRP. However, existing multi-task models for VRP variants predominantly rely on heavy encoder-light decoder architectures, exhibiting poor scale generalization. To address this limitation, as depicted in Figure 2, we have developed a multi-task heavy decoder model specifically designed for VRP variants, aiming to overcome the inadequate generalization capabilities of current approaches.

## 4.1 Encoder

Given an instance $S$ of the VRP, which includes $N$ node features $(\mathbf{s}_1, ..., \mathbf{s}_n)$, in VRP variants, where each node feature $\mathbf{s}_i$ is represented as $(\mathbf{x}_i, d_i, \delta_i, s_i, l_i)$, denoting the coordinate information, demand, service time, and the start and end times of the time window, respectively. These node features are then mapped through a linear layer to obtain the initial embedding matrix $H^{init} = (\mathbf{h}_1^{init}, ..., \mathbf{h}_n^{init})$, where $\mathbf{h}_i^{init} = \text{Linear}(\mathbf{s}_i)$, and $\text{Linear}(\cdot)$ denotes the linear layer. Subsequently, these initial embeddings are passed through a Transformer layer to capture node relationships and generate the node embeddings. Therefore, the output of the encoder can be represented as:

$$H^{enc} = \text{TFL}(H^{init}) = \text{TFL}(\mathbf{h}_1^{init}, ..., \mathbf{h}_n^{init}),$$

where $H^{enc} = (\mathbf{h}_1^{enc}, ..., \mathbf{h}_n^{enc})$ is the matrix of final node embeddings, $\text{TFL}(\cdot)$ denote the Transformer layer, and $\mathbf{h}_i$ represents the embedding of the $i$-th node.

## 4.2 Decoder

At the step $t$ during the decoding phase, we first extract the dynamic features $D = \{l_r, t_c, d_r, o\}$, encompassing the remaining vehicle load, current time, remaining route duration, and a binary flag indicating whether the route is open. These dynamic features are combined with the last visited node $(\mathbf{h}_{last})$ and the depot node $(\mathbf{h}_{depot})$, respectively, and then each combination is passed through its own linear layer. The resulting embeddings, along with the unvisited nodes' embeddings $(\mathbf{h}_{unvisited})$, are processed by an $L$-layer Transformer network to yield updated node embeddings:

$$
\begin{aligned}
H^{(0)} &= \text{concat}(\mathbf{h}_{unvisited}^{enc}, \text{Linear}(D \oplus \mathbf{h}_{last}^{enc}), \text{Linear}(D \oplus \mathbf{h}_{depot}^{enc})), \\
H^{(1)} &= \text{TFL}(H^{(0)}; M^{\text{pad}}), \\
&\cdots \\
H^{(L)} &= \text{TFL}(H^{(L-1)}; M^{\text{pad}}).
\end{aligned}
\tag{1}
$$

Here, $M^{\text{pad}} \in \{0, -\infty\}^N$ denotes the padding mask, where 0 marks valid positions and $-\infty$ marks padded positions. We introduce this mask to handle different numbers of unvisited nodes across batch instances (which affects efficiency): sequences are padded to a unified length by appending depot tokens to shorter sequences and then masked by $M^{\text{pad}}$.

Next, we compute compatibility scores between the embeddings of all unvisited nodes and a context embedding $\mathbf{h}_q = \text{Linear}(\text{concat}(\mathbf{h}_{last}^{(L)}, \mathbf{h}_{depot}^{(L)}))$ using single-head attention:

$$c(\mathbf{h}_i^{(L)}, \mathbf{h}_q) = \text{SHA}(\mathbf{h}_i^{(L)}, \mathbf{h}_q).$$

Before selection, we add a padding mask and a feasibility mask to the logits to ensure valid routes. Finally, the probability $\pi(i|s_t)$ of selecting the $i$-th unvisited node is determined by the softmax function applied to the masked compatibility scores:

$$\pi(i|s_t) = \text{softmax}\big(c(\mathbf{h}_i^{(L)}, \mathbf{h}_q) + M_i^{\text{pad}} + M_i^{\text{feas}}\big),$$

where $c(\mathbf{h}_i^{(L)}, \mathbf{h}_q)$ is the compatibility score from SHA, $\mathbf{h}_i^{(L)}$ is the embedding of the $i$-th unvisited node, and $s_t$ is the current decoder state. The masks take values in $\{0, -\infty\}$: $M^{\text{pad}}$ excludes padded tokens and $M^{\text{feas}}$ excludes currently infeasible moves.

It is noteworthy that we remove layer normalization from all attention layers [27].

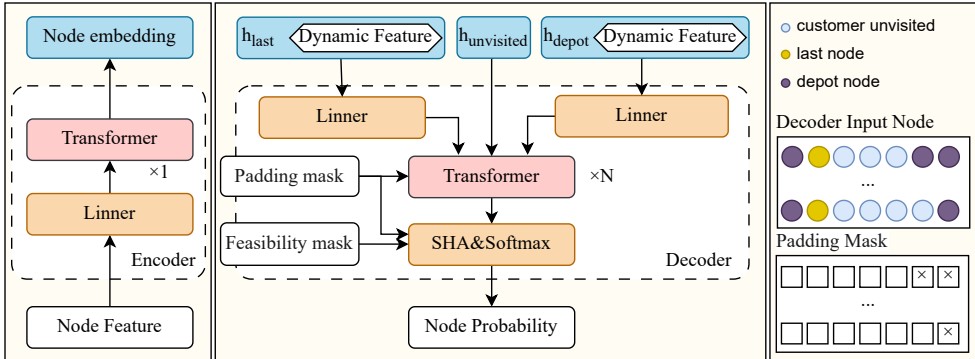

Figure 2: Architecture of the proposed multi-task neural solver. Left to right: Encoder, Decoder, Node Padding & Masking.

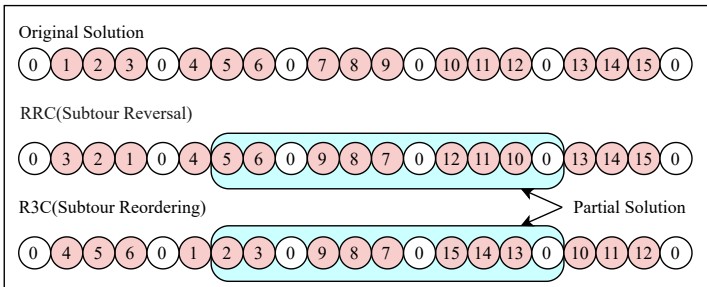

Figure 3: Comparison of RRC and R3C methods. RRC increases the diversity of sampled subproblems by randomly reversing subtours, while R3C enhances this diversity by randomly reordering the external sequence of subtours.

## 5 Random Reordering Re-Construct

Random Re-Construct (RRC) [27] is an iterative method to improve solution quality by randomly sampling and re-optimizing segments of the initial solution. RRC enhances sampling diversity via random subtour reversals and segment lengths. However, reversals can violate feasibility for some problems (e.g., VRPTW), and sampling solely on the original sequence limits diversity, hindering iterative performance. We thus propose Random Reordering Re-Construct to increase sampling diversity and improve iterative performance.

The R3C strategy is detailed in Figure 3. Given a solution, we first decompose it into subtours, then randomize their external order in the sequence. Subsequently, a random-length contiguous segment is randomly sampled and re-optimized by our model. If the re-optimized segment improves the objective value, it replaces the original segment. By randomizing subtour order, R3C allows for more random combinations of subtours into partial solutions, aiding escape from local optima and improving iterative performance. All sampled segments end at the depot. Additionally, feasible subtour reversals are also incorporated to further enhance search diversity.

## 6 Experiments

In this section, we present a series of experiments conducted on 16 VRP variants to validate the performance of our model. All experiments are performed using one NVIDIA RTX 4090 GPU.

**Baselines** The baseline methods are categorized into two groups: traditional heuristic solvers and multi-task neural solvers. **(1) Traditional Solvers:** PyVRP [36] is an open-source Hybrid Genetic

Search (HGS)-based solver supporting multiple complex VRP variants. It represents the current state-of-the-art in heuristic approaches for the problems considered in this study. OR-Tools [9] is Google's open-source optimization toolkit, applicable to all VRP variants examined in this study.

**(2) Multi-task Neural Solvers:** MT-POMO [23] is a multi-task extension of the POMO framework, enabling simultaneous learning across different VRP variants. MVMoE [24] is an architecture that enhances model capacity and improves performance over MT-POMO by incorporating Mixture-of-Experts (MoE) modules. RouteFinder [25] is a multi-task training framework that enhances training efficiency through mixed batch training and enables fast adaptation to unseen tasks via lightweight adapter layers. The framework includes three model variants: RF-Transformer, RF-POMO, and RF-MVMoE. CaDA [37] incorporates constraint awareness and a dual-branch structure to better handle diverse tasks within a multi-task learning framework.

**Datasets**   We adopt the same problem settings as MVMoE [24]. We generate a test set comprising 16 VRP problems across four scales: 100, 200, 500, and 1000. The dataset at the 100-scale contains 1,000 instances, while the others contain 128 instances each. In these settings, the vehicle capacity is uniformly set to 50, the duration limit is set to 3, the time window for the depot node in Time Window problems is [0, 3.0], and for backhaul problems, the proportion of backhaul nodes is set to 20%. Our datasets are generated based on the MVMoE codebase to evaluate the performance of various baselines.

**Training Configuration and Model Parameters**   We follow a similar setup to MVMoE, training our model on 6 VRPs and evaluating its zero-shot generalization performance on 10 unseen tasks (refer to Figure 1). The training process consists of two phases: teacher model pre-training and student model training.**(1) Teacher Model Pre-training:** We employ the POMO model as our teacher model. For each of the 6 visible tasks, we independently train a single-task model using reinforcement learning. All teacher models share the same parameter configuration: 6 encoder layers, 1 decoder layer, an embedding dimension of 128, 8 attention heads, and a Feedforward hidden dimension of 512. The teacher models are trained for 4000 epochs on random instances of scale 100, with 20,000 training instances per epoch, a batch size of 128, and an initial learning rate of 0.0001. **(2) Student Model Training:** The student model consists of 1 encoder layer and 6 decoder layers, with embedding dimensions set to 128 and 96, employing 8 multi-head attention heads and a Feedforward hidden dimension of 512. The student model simultaneously learns from the 6 pre-trained teacher models on datasets with a problem scale of 100. The training batch size is set to 1500 ($250 \times 6$), with 24,000 training instances per epoch, an initial learning rate of 0.0001, and a total of 850 training epochs. After the 300th epoch, the learning rate is halved every 100 epochs.

**Inference and Metrics**   We adopt the Average Objective Value and Gap as evaluation metrics, where smaller values indicate better performance. The Objective Value represents the total distance of the solution in the Vehicle Routing Problem, while the Gap evaluates the performance difference of each method compared to a traditional baseline method (such as HGS-PyVRP).

We test all methods using Single Trajectory Greedy Search (S.T.). Furthermore, to compare with the results of other solvers exploring multiple trajectories under data augmentation (denoted as M×aug8, where M is the number of nodes and aug8 represents 8 types of data augmentation [11]), our method employs the R3C strategy with 200 iterations under a single trajectory.

## 6.1   Main Results

**Performance on seen Tasks**   We evaluate the performance of our model on the training tasks, and the results are shown in Table 1. The results demonstrate that our proposed MTL-KD model exhibits excellent performance on the seen training tasks, particularly showcasing a more pronounced advantage when dealing with larger-scale problems. This validates the effectiveness of our proposed knowledge distillation training framework and model architecture.

**Performance on Unseen Tasks**   The zero-shot generalization experimental results on 10 unseen tasks (as shown in Table 2) indicate that our MTL-KD model outperforms the compared models in most cases, demonstrating remarkable cross-task generalization ability, especially with a significant advantage on large-scale problems. This validates that multi-task knowledge distillation training endows the model with strong generalization capabilities across different VRP variants.

Table 1: Performance on seen VRPs across three scales. * represents the baseline used for gap calculation.

| Method | Pro. | Problem Size | | | Pro. | Problem Size | | |
|---|---|---|---|---|---|---|---|---|
| | | n=100 | n=500 | n=1k | | n=100 | n=500 | n=1k |
| HGS-PyVRP | | 15.53(*) | 62.07(*) | 119.54(*) | | 24.35(*) | 90.61(*) | 166.47(*) |
| OR-Tools | | 15.94(2.63%) | 66.51(7.15%) | 125.91(5.32%) | | 25.21(3.54%) | 98.45(8.65%) | 178.47(7.21%) |
| MT-POMO(S.T.) | | 16.25(4.64%) | 70.60(13.74%) | 146.92(22.90%) | | 26.66(10.40%) | 122.61(35.50%) | 247.44(66.47%) |
| MT-POMO(M+aug8) | | 15.79(1.69%) | 67.99(9.54%) | 136.62(14.28%) | | 25.61(5.18%) | 115.43(27.39%) | 229.82(38.06%) |
| MVMoE(S.T.) | | 16.30(5.00%) | 77.44(24.77%) | 191.07(59.83%) | | 26.88(10.44%) | 122.78(35.26%) | 267.33(60.59%) |
| MVMoE(M+aug8) | CVRP | 15.76(1.50%) | 73.61(18.59%) | 176.40(47.57%) | VRPTW | 25.51(4.78%) | 116.67(28.76%) | 253.35(52.19%) |
| RF-Transformer(M+aug8) | | 15.82(1.88%) | 67.71(9.08%) | 132.79(11.08%) | | 27.39(12.49%) | 108.70(19.97%) | 222.92(33.91%) |
| RF-MTPOMO(M+aug8) | | 15.87(2.20%) | 67.42(8.62%) | 132.82(11.11%) | | 26.29(7.98%) | 102.77(13.42%) | 193.57(16.28%) |
| RF-MVMoE(M+aug8) | | 15.84(2.01%) | 67.36(8.52%) | 134.85(12.80%) | | 26.29(7.98%) | 100.78(11.22%) | 187.87(12.86%) |
| CaDa(M+aug8) | | 15.84(2.00%) | 175.65(182.99%) | 542.56(353.87%) | | 30.16(23.87%) | 300.11(231.21%) | 693.61(316.66%) |
| MTL-KD$_{96}$ | | 16.06(3.41%) | 64.57(4.02%) | 123.88(3.63%) | | 26.18(7.52%) | 100.17(10.55%) | 188.94(13.50%) |
| MTL-KD$_{128}$ | | 16.04(3.30%) | 64.61(4.09%) | 124.44(4.09%) | | 26.13(7.32%) | 99.05(9.31%) | 184.92(11.09%) |
| MTL-KD(R3C200)$_{128}$ | | **15.76(1.48%)** | **63.63(2.51%)** | **122.06(2.10%)** | | **25.31(3.93%)** | **96.43(6.42%)** | **181.85(9.24%)** |
| HGS-PyVRP | | 15.58(*) | 63.55(*) | 122.68(*) | | 9.71(*) | 35.30(*) | 66.10(*) |
| OR-Tools | | 16.00(2.69%) | 67.53(6.26%) | 128.16(4.46%) | | 9.84(1.38%) | 37.81(7.11%) | 70.38(6.48%) |
| MT-POMO(S.T.) | | 16.29(4.52%) | 71.46(12.66%) | 149.5521.90(%) | | 10.66(9.83%) | 44.62(26.41%) | 92.78(40.36%) |
| MT-POMO(M+aug8) | | 15.85(1.67%) | 68.95(8.49%) | 138.97(13.27%) | | 10.17(4.74%) | 41.93(18.78%) | 85.27(29.00%) |
| MVMoE(S.T.) | | 16.36(4.94%) | 78.79(23.97%) | 192.29(56.74%) | | 10.77(10.94%) | 49.35(39.81%) | 135.30(104.70%) |
| MVMoE(M+aug8) | VRPL | **15.81(1.45%)** | 74.67(17.50%) | 177.89(45.00%) | OVRP | 10.14(4.42%) | 46.23(30.97%) | 116.82(76.74%) |
| RF-Transformer(M+aug8) | | 15.88(1.93%) | 68.59(7.93%) | 135.15(10.17%) | | 10.11(4.11%) | 42.88(21.48%) | 84.40(27.68%) |
| RF-MTPOMO(M+aug8) | | 15.93(2.23%) | 68.33(7.52%) | 134.53(9.66%) | | 10.17(4.70%) | 41.60(17.86%) | 82.14(24.26%) |
| RF-MVMoE(M+aug8) | | 15.90(2.06%) | 68.39(7.62%) | 138.60(12.98%) | | 10.13(4.29%) | 41.18(16.65%) | 81.82(23.78%) |
| CaDa(M+aug8) | | 15.89(2.00%) | 176.04(177.01%) | 536.71(337.49%) | | 10.10(4.04%) | 187.72(431.78%) | 442.57(569.55%) |
| MTL-KD$_{96}$ | | 16.14(3.58%) | 65.45(2.99%) | 126.11(2.79%) | | 10.41(7.21%) | 38.73(9.72%) | 72.73(10.03%) |
| MTL-KD$_{128}$ | | 16.12(3.45%) | 65.48(3.03%) | 126.66(3.25%) | | 10.46(7.19%) | 38.89(10.17%) | 72.70(9.98%) |
| MTL-KD(R3C200)$_{128}$ | | 15.82(1.50%) | **64.5246(1.53%)** | **124.59(1.56%)** | | **10.05(3.53%)** | **37.7934(7.07%)** | **71.40(8.03%)** |
| HGS-PyVRP | | - | - | - | | 13.95(*) | 48.15(*) | 82.98(*) |
| OR-Tools | | 11.97(*) | 47.76(*) | 88.57(*) | | 14.38(3.06%) | 52.48(9.00%) | 90.03(8.49%) |
| MT-POMO(S.T.) | | 12.47(4.21%) | 50.08(4.85%) | 99.56(12.41%) | | 15.56(11.53%) | 71.63(48.76%) | 147.84(78.16%) |
| MT-POMO(M+aug8) | | 12.04(0.63%) | 48.49(1.52%) | 95.35(7.64%) | | 14.85(6.43%) | 66.74(38.61%) | 135.99(63.87%) |
| MVMoE(S.T.) | VRPB | 12.42(3.80%) | 71.99(50.75%) | 186.49(110.54%) | OVRPTW | 15.70(12.54%) | 82.81(71.99%) | 215.21(159.34%) |
| MVMoE(M+aug8) | | **12.01(0.31%)** | 66.26(38.73%) | 167.26(88.84%) | | 14.78(5.90%) | 76.28(58.43%) | 195.64(135.76%) |
| MTL-KD$_{96}$ | | 12.39(3.51%) | 46.12(-3.44%) | 86.90(-1.89%) | | 15.11(8.32%) | 53.74(11.61%) | 94.12(13.42%) |
| MTL-KD$_{128}$ | | 12.38(3.41%) | 45.99(-3.70%) | 86.99(-1.78%) | | 15.08(8.07%) | 53.89(11.91%) | 93.96(13.22%) |
| MTL-KD(R3C200)$_{128}$ | | 12.02(0.43%) | **44.58(-6.66%)** | **83.66(-5.55%)** | | **14.53(4.12%)** | **52.08(8.15%)** | **92.40(11.35%)** |

**Performance on Real-World Instances**   We also evaluate the performance of our model on real-world datasets, including Set-X (medium and large scale) from CVRPLIB [38] for CVRP and the Solomon dataset [39] for VRPTW. We compare our model against several single-task and multi-task models, and the experimental results are presented in Tables 6, 7, and 8, AppendixD. Our model consistently outperforms other single-task and multi-task models across all three real-world datasets, demonstrating particularly strong performance on the large-scale Set-X dataset, where the gap is significantly smaller than other baselines. This indicates the strong applicability of our approach to real-world scenarios.

## 6.2   Ablation Study

**Scale Generalization Comparison: Student vs. Teacher**   While heavy decoders are generally considered to have significant potential for large-scale generalization, the teacher model, POMO, primarily excels at its specific training scale and lacks cross-scale generalization capabilities. This naturally raises a crucial question: Will the student model also be limited to learning knowledge specific to a particular scale, thereby restricting its generalization ability? To investigate this, we compare the performance of several single-task teacher models with our MTL-KD model on the same dataset, as shown in Figure 4. In contrast to the teacher models, the student model demonstrates a notable capacity for scale generalization. This observation indicates that the MTL-KD model, during the learning process, does not simply replicate the teacher's knowledge but rather effectively adapts to scale variations, consequently achieving robust generalization performance.

**Performance Comparison: KD vs. RL**   To highlight the effectiveness of our knowledge distillation approach, we train the proposed multi-task LEHD model under two distinct training paradigms: knowledge distillation (MTL-KD) and reinforcement learning (MTL-RL). Given the high computational cost associated with training the LEHD model with RL, we are only able to train it at a scale of 20. Subsequently, we evaluate the average gap of both models in all tasks seen and unseen , with the experimental results presented in Table 3. The results indicate that the RL method, due to its limitation to training on small scales, struggles to leverage the full potential of the LEHD model,

Table 2: Performance on unseen VRPs across three scales. * represents the baseline used for gap calculation.

| Method | Pro. | Problem Size | | | Pro. | Problem Size | | |
|---|---|---|---|---|---|---|---|---|
| | | n=100 | n=500 | n=1k | | n=100 | n=500 | n=1k |
| HGS-PyVRP | OVRPL | 9.67(*) | 34.70(*) | 65.38(*) | VRPLTW | 24.44(*) | 91.86(*) | 174.79(*) |
| OR-Tools | | 9.79(1.25%) | 37.09(6.89%) | 69.64(6.51%) | | 25.65(4.96%) | 99.50(8.31%) | 186.38(6.63%) |
| MT-POMO(S.T.) | | 10.61(9.69%) | 43.77(26.13%) | 92.49(41.45%) | | 26.76(9.52%) | 123.97(34.95%) | 252.82(44.65%) |
| MT-POMO(M+aug8) | | 10.13(4.72%) | 41.28(18.95%) | 84.40(29.08%) | | 25.71(5.21%) | 116.62(26.95%) | 235.94(34.99%) |
| MVMoE(S.T.) | | 10.76(11.25%) | 48.60(40.05%) | 135.08(106.58%) | | 27.00(10.48%) | 124.25(35.25%) | 274.68(57.15%) |
| MVMoE(M+aug8) | | 10.10(4.43%) | 45.58(31.36%) | 116.71(78.48%) | | 25.62(4.84%) | 118.00(28.45%) | 260.07(48.79%) |
| MTL-KD(S.T.)$_{96}$ | | 10.42(7.73%) | 38.23(10.17%) | 72.32(10.61%) | | 26.37(7.89%) | 101.39(10.36%) | 195.65(11.94%) |
| MTL-KD(S.T.)$_{128}$ | | 10.44(7.95%) | 38.43(10.76%) | 72.78(11.31%) | | 26.35(7.81%) | 100.40(9.30%) | 192.41(10.08%) |
| MTL-KD(R3C200)$_{128}$ | | 10.08(4.29%) | 37.27(7.41%) | 71.35(9.12%) | | 25.55(4.54%) | 97.85(6.51%) | 189.02(8.14%) |
| HGS-PyVRP | OVRPLTW | 14.00(*) | 47.97(*) | 83.68(*) | OVRPB | - | - | - |
| OR-Tools | | 14.28(1.98%) | 52.94(10.36%) | 91.69(9.57%) | | 8.37(*) | 29.98(*) | 54.87(*) |
| MT-POMO(S.T.) | | 15.61(11.46%) | 71.24(48.52%) | 147.97(76.82%) | | 9.52(13.82%) | 35.43(18.17%) | 74.59(35.95%) |
| MT-POMO(M+aug8) | | 14.90(6.39%) | 66.65(38.95%) | 137.13(63.87%) | | 8.98(7.34%) | 32.76(9.28%) | 66.91(21.95%) |
| MVMoE(S.T.) | | 15.74(12.39%) | 82.49(71.97%) | 219.21(161.95%) | | 9.74(16.41%) | 45.80(52.76%) | 129.38(135.81%) |
| MVMoE(M+aug8) | | 14.83(5.90%) | 76.12(58.68%) | 197.84(136.41%) | | 8.86(7.10%) | 40.41(34.77%) | 109.08(98.81%) |
| MTL-KD(S.T.)$_{96}$ | | 15.28(9.12%) | 53.38(11.28%) | 94.65(13.11%) | | 9.25(10.58%) | 30.90(3.08%) | 58.57(6.76%) |
| MTL-KD(S.T.)$_{128}$ | | 15.26(8.95%) | 53.56(11.65%) | 95.33(13.92%) | | 9.27(10.81%) | 30.45(1.58%) | 56.92(3.73%) |
| MTL-KD(R3C200)$_{128}$ | | 14.71(5.02%) | 51.96(8.34%) | 93.91(12.22%) | | 8.78(4.95%) | 28.73(-4.16%) | 52.41(-4.48%) |
| OR-Tools | VRPBL | 12.02(*) | 47.93(*) | 89.82(*) | VRPBTW | 25.41(*) | 97.77(*) | 194.69(*) |
| MT-POMO(S.T.) | | 12.71(5.79%) | 50.70(5.77%) | 99.51(10.78%) | | 28.64(12.70%) | 126.00(28.87%) | 270.68(39.03%) |
| MT-POMO(M+aug8) | | 12.10(0.66%) | 48.14(0.45%) | 94.26(4.94%) | | 26.94(6.04%) | 118.20(20.89%) | 251.58(29.22%) |
| MVMoE(S.T.) | | 12.86(7.06%) | 70.03(46.11%) | 159.91(78.03%) | | 28.88(13.67%) | 125.89(28.76%) | 294.23(51.13%) |
| MVMoE(M+aug8) | | 12.05(0.28%) | 63.57(32.63%) | 145.61(62.10%) | | 26.89(5.82%) | 119.44(22.16%) | 276.17(41.85%) |
| MTL-KD(S.T.)$_{96}$ | | 12.56(4.54%) | 45.90(-4.23%) | 86.45(-3.75%) | | 28.48(12.11%) | 105.43(7.83%) | 215.94(10.92%) |
| MTL-KD(S.T.)$_{128}$ | | 12.54(4.32%) | 45.67(-4.72%) | 86.28(-3.95%) | | 28.49(12.12%) | 103.85(6.21%) | 209.82(7.77%) |
| MTL-KD(R3C200)$_{128}$ | | 12.08(0.55%) | 44.27(-7.64%) | 82.56(-8.08%) | | 26.64(4.84%) | 100.25(2.53%) | 205.13(5.37%) |
| OR-Tools | OVRPBL | 8.35(*) | 29.60(*) | 54.30(*) | OVRPBTW | 14.38(*) | 51.88(*) | 90.86(*) |
| MT-POMO(S.T.) | | 9.50(13.79%) | 35.34(19.37%) | 74.83(37.80%) | | 16.92(17.64%) | 72.91(40.53%) | 152.63(67.98%) |
| MT-POMO(M+aug8) | | 8.96(7.35%) | 32.60(10.13%) | 66.87(23.13%) | | 15.88(10.39%) | 67.95(30.96%) | 140.72(54.87%) |
| MVMoE(S.T.) | | 9.75(16.75%) | 45.46(53.57%) | 127.34(134.49%) | | 17.07(18.68%) | 81.57(57.22%) | 214.76(136.36%) |
| MVMoE(M+aug8) | | 8.94(7.12%) | 40.35(36.29%) | 108.53(99.86%) | | 15.81(9.90%) | 74.43(43.46%) | 193.33(112.77%) |
| MTL-KD(S.T.)$_{96}$ | | 9.29(11.27%) | 31.96(7.96%) | 61.37(13.01%) | | 16.70(16.10%) | 56.02(7.97%) | 99.85(9.89%) |
| MTL-KD(S.T.)$_{128}$ | | 9.41(12.75%) | 31.42(6.14%) | 59.25(9.11%) | | 16.74(16.39%) | 56.02(7.98%) | 99.20(9.18%) |
| MTL-KD(R3C200)$_{128}$ | | 8.84(5.95%) | 29.23(-1.26%) | 53.77(-0.98%) | | 15.58(8.30%) | 53.75(3.62%) | 97.37(7.17%) |
| OR-Tools | VRPBLTW | 25.34(*) | 103.17(*) | 189.31(*) | OVRPBLTW | 14.25(*) | 52.37(*) | 92.16(*) |
| MT-POMO(S.T.) | | 28.92(14.11%) | 131.68(27.64%) | 264.39(39.66%) | | 16.78(17.79%) | 73.26(39.90%) | 152.61(65.60%) |
| MT-POMO(M+aug8) | | 27.25(7.52%) | 123.71(19.91%) | 245.61(29.74%) | | 15.74(10.45%) | 68.08(30.01%) | 141.27(53.28%) |
| MVMoE(S.T.) | | 29.19(15.18%) | 132.23(28.17%) | 286.27(51.21%) | | 16.95(18.94%) | 82.36(57.28%) | 215.69(134.03%) |
| MVMoE(M+aug8) | | 27.14(7.10%) | 125.27(21.43%) | 269.13(42.16%) | | 15.67(9.97%) | 74.83(42.90%) | 195.45(112.77%) |
| MTL-KD(S.T.)$_{96}$ | | 29.01(14.4703%) | 111.33(7.92%) | 210.45(11.17%) | | 16.78(17.74%) | 56.34(7.59%) | 101.43(10.06%) |
| MTL-KD(S.T.)$_{128}$ | | 29.13(14.96%) | 110.33(6.94%) | 205.52(8.56%) | | 16.95(18.93%) | 57.15(9.13%) | 101.64(10.29%) |
| MTL-KD(R3C200)$_{128}$ | | 27.12(7.03%) | 106.52(3.26%) | 201.29(6.33%) | | 15.65(9.84%) | 54.68(4.41%) | 99.64(8.11%) |

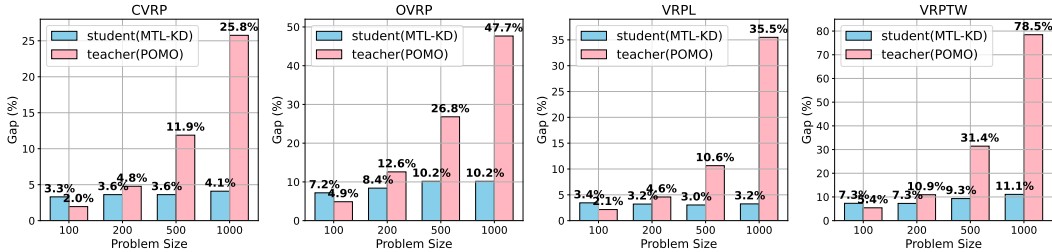

Figure 4: Performance Comparison between Teacher and Student Models, both trained on instances of scale 100.

leading to significantly poorer performance. In contrast, our distillation-based MTL-KD model demonstrates a clear performance advantage.

**Effectiveness Analysis of R3C**    To validate the effectiveness of our proposed R3C method, we conduct experiments on CVRP, VRPL, OVRP, and VRPTW at scale 100. We perform an ablation study on the random reordering of the subtour external order component, with the following comparative experiments: random sampling on the original solution only (RS); random reordering of the subtour external order followed by sampling (RS+Ro); and for CVRP and VRPL, we also analyze the impact of adding a random flipping operation (F) since it does not affect the legality of the solution. The experimental results are shown in Figure 5. Incorporating the random reordering operation significantly improves the iterative performance during reconstruction, which benefits from the ability of random reordering to enhance the diversity of sampled subproblems. Furthermore, the random flipping operation can moderately increase the diversity of sampled solutions when

Table 3: Heavy Decoder Performance (RL vs. KD) on Seen/Unseen Tasks (Average Gap).

| Problem Size | | 100 | 200 | 500 | 1000 |
|---|---|---|---|---|---|
| Training task | MTL-RL | 21.7834% | 27.2457% | 38.2808% | 50.0606% |
| | MTL-KD | **5.4569%** | **5.2775%** | **5.8028%** | **6.6413%** |
| unseen task | MTL-RL | 31.2513% | 37.1404% | 48.9074% | 62.2226% |
| | MTL-KD | **19.1658%** | **12.9145%** | **10.8275%** | **13.3322%** |

only initial solution random sampling is performed; however, its impact is minimal with random reordering, further confirming the effectiveness of the random reordering operation.

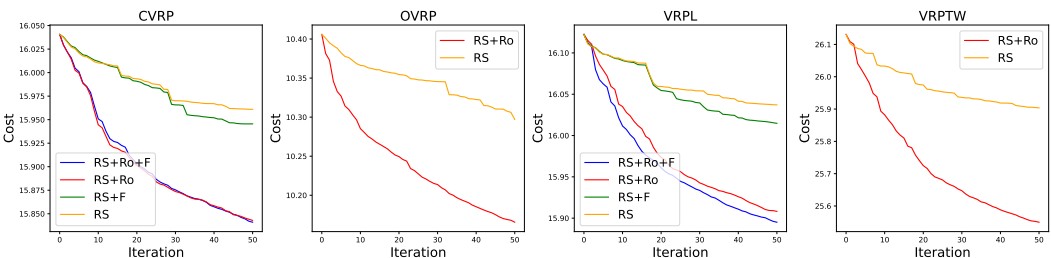

Figure 5: Impact of Different Components in R3C.

## 7 Conclusion

This paper presents a heavy decoder model for the multi-task VRP domain and achieves label-free training of this model through a multi-task knowledge distillation method. The proposed approach has demonstrated excellent performance on both seen and unseen tasks, as well as real-world datasets, exhibiting significant large-scale generalization ability and thereby validating the effectiveness of our method. Furthermore, the proposed R3C method has further enhanced the model's performance. However, the structure of the heavy decoder results in high computational complexity. Future work will explore how to implement more efficient and robust model architectures in the multi-task domain.

## Acknowledgments and Disclosure of Funding

This work was supported in part by the National Natural Science Foundation of China (Grant 62476118, Grant 72271168), the Natural Science Foundation of Guangdong Province (Grant 2024A1515011759), the Guangdong Science and Technology Program (Grant 2024B1212010002), the Guangdong Basic and Applied Basic Research Foundation (Grant 2024A1515012485), and the Center for Computational Science and Engineering at Southern University of Science and Technology.

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

# A Related work

## A.1 Single-task Autoregressive Neural Solver

Research on high-performance autoregressive neural solvers for VRP is a hot topic in NCO. It is typically based on the encoder-decoder structure, where the encoder generates node embeddings, and the decoder is used to dynamically generate node sequences. Early works are based on Pointer Networks [40], gradually evolving from supervised learning to reinforcement learning training [41, 42]. Subsequently, the Transformer framework is introduced and becomes a mainstream paradigm [10]. POMO [11] further leverages the symmetry of the VRP, optimizing the baseline calculation in RL training and enhancing the exploration diversity of the model. Subsequent research largely focuses on improving the Transformer framework [43, 44, 13, 12, 45–47]. However, these approaches predominantly adopt the Heavy Encoder and Light Decoder (HELD) structure, which poses challenges in handling large-scale problems. To enhance the capability of solving large instances, some methods continue to build upon HELD. Examples include ELG [48], which integrates local search strategies; INViT [49], which utilizes K-nearest neighbor multi-view embeddings to enhance local information; and ICAM [50], which introduces instance-conditional adaptation to better perceive problem scales. On the other hand, some studies [27, 28] explore heavy decoder architectures. By dynamically capturing the relationships between remaining nodes during the decoding process, these methods demonstrate superior generalization performance on large-scale problems. However, their high computational demands, often necessitating training on labeled data, represent a significant limitation. Furthermore, some methods [51–53, 29] decompose large-scale problems into multiple smaller subproblems for solving, and autoregressive neural solvers can be employed at the lower level to address these subproblems.

## A.2 Multi-Task Autoregressive Neural Solver

To enhance the generalizability of neural solvers across diverse VRP variants, several existing methods have proposed unified multi-task models. For example, Wang and Yu [54] employ a multi-armed bandit approach to achieve efficient training across multiple combinatorial optimization problems. Lin et al.[55] pre-train a Transformer backbone on the Traveling Salesman Problem and fine-tune it for specific VRP variants, thereby extending its applicability to a subset of VRP problems. These methods represent initial attempts at cross-problem learning but are limited in scope, focusing primarily on a small number of problems. Building on these foundations, MT-POMO [23] views VRP variants as combinations of distinct attributes, enabling unified representation and zero-shot generalization across multiple VRP variants, achieving notable performance on ten tasks. MVMoE [24] further boosts model capacity and multi-task performance by introducing a mixture-of-experts model. RouteFinder [25] enhances multi-task training efficiency and performance through mixed batch training and enables rapid adaptation to unseen tasks via efficient adapter layers. CaDA [37] improves the model's ability to handle diverse tasks within a multi-task framework by incorporating constraint awareness and a dual-branch structure. However, these studies have predominantly focused on small-scale problems, lacking extensive exploration of large-scale generalization capabilities.

# B Problem Definition and Settings

Our VRP variant problem settings follow the work of Zhou et al. [24]. The detailed settings are as follows:

- **Coordinates**: All node coordinates are uniformly sampled from the 2D space $[0, 1) \times [0, 1)$.

- **Demands**: For all customer nodes, demands are randomly sampled from the set $\{1, 2, \ldots, 9\}$.

- **Vehicle Capacity**: Each vehicle has a fixed transportation capacity, which we set to 50 across all problem scales. The sum of demands for nodes visited by a vehicle must be less than or equal to its current capacity.

- **Open Routes**: In open vehicle routing problems, vehicles are not required to return to the depot. We use a binary flag vector to distinguish whether a path is open; setting it to 1 indicates an open path.

Table 4: 16 VRP variants with five constraints.

| Constraint | Capacity | Open Route | Backhaul | Duration Limit | Time Window |
|---|---|---|---|---|---|
| CVRP | ✓ | | | | |
| OVRP | ✓ | ✓ | | | |
| VRPB | ✓ | | ✓ | | |
| VRPL | ✓ | | | ✓ | |
| VRPTW | ✓ | | | | ✓ |
| OVRPTW | ✓ | ✓ | | | ✓ |
| OVRPB | ✓ | ✓ | ✓ | | |
| OVRPL | ✓ | ✓ | | ✓ | |
| VRPBL | ✓ | | ✓ | ✓ | |
| VRPBTW | ✓ | | ✓ | | ✓ |
| VRPLTW | ✓ | | | ✓ | ✓ |
| OVRPBL | ✓ | ✓ | ✓ | ✓ | |
| OVRPBTW | ✓ | ✓ | ✓ | | ✓ |
| OVRPLTW | ✓ | ✓ | | ✓ | ✓ |
| VRPBLTW | ✓ | | ✓ | ✓ | ✓ |
| OVRPBLTW | ✓ | ✓ | ✓ | ✓ | ✓ |

- **Backhauls**: In problems involving backhauls, a portion of the nodes have negative demands, referred to as backhaul nodes. We designate 20% of the customer nodes as backhaul nodes, whose demands are randomly sampled from $\{-1, -2, \ldots, -9\}$.
- **Duration Limit**: Each vehicle has a maximum travel distance limit, which we set to 3.
- **Time Windows**: Customer node time windows are uniformly sampled, and service time is uniformly set to 0.2. Vehicles must visit nodes within their time windows; if a vehicle arrives early, it must wait until the earliest start time. The depot's time window is set to $[0, 3]$, with a service time of 0. Vehicle speed is uniformly set to 1, meaning the time spent traveling on a path equals its distance divided by speed. Notably, for non-open problems, it is crucial to ensure that after visiting the last customer node, the vehicle can return to the depot within its latest allowable time window; this consideration is not necessary for open problems.

By combining different constraints, 16 VRP variants can be formed, as detailed in Table 4.

## C Sensitivity Analysis on R3C Re-optimization Segment Length

To confirm the efficacy of our random segment length selection in the R3C strategy, we compare it against fixed segment lengths ($k \in \{10, 20, 30, 40, 50\}$) on 100-node instances for four VRP variants (CVRP, OVRP, VRPL, VRPTW). As shown in Table 5, the random length consistently yields the best performance across all problems. We attribute this advantage to the increased solution diversity provided by varying segment lengths, which effectively prevents the search from getting trapped in local optima, a common issue with fixed segment sampling.

Table 5: Results after 50 iterations of R3C with different Re-optimization Segment Lengths on 100-node instances.

| Problem | segment=10 | segment=20 | segment=30 | segment=40 | segment=50 | **Random** |
|---|---|---|---|---|---|---|
| CVRP | 16.01 | 15.96 | 15.94 | 15.91 | 15.91 | **15.88** |
| OVRP | 10.34 | 10.25 | 10.21 | 10.17 | 10.20 | **10.17** |
| VRPL | 16.09 | 16.03 | 16.00 | 15.97 | 15.98 | **15.90** |
| VRPTW | 26.02 | 25.86 | 25.77 | 25.65 | 25.69 | **25.55** |

## D Performance on Real-World Instances

Our test results on real-world instance datasets are presented in Tables 6, 7, and 8. All models are trained on instances of size $N = 100$. During inference, data augmentation is applied, and a

greedy rollout strategy is adopted by default. Some results for comparative models are sourced from MVMoE [24] and RouteFinder [25].

Table 6: Results on small-scale CVRPLIB instances (Set-X) [38]. All models are trained on instances of size $N = 100$, and inference utilizes data augmentation [11] and greedy rollout by default.

| Set-X | | POMO | | POMO-MTL | | MVMoE/4E | | MVMOE/4E-L | | MTL-KD | |
|---|---|---|---|---|---|---|---|---|---|---|---|
| Instance | Opt. | Obj. | Gap | Obj. | Gap | Obj. | Gap | Obj. | Gap | Obj. | Gap |
| X-n101-k25 | 27591 | 30138 | 9.231% | 32482 | 17.727% | 29361 | 6.415% | **29015** | **5.161%** | 29058 | 5.317% |
| X-n106-k14 | 26362 | 39322 | 49.162% | 27369 | 3.820% | 27278 | 3.475% | 27242 | 3.338% | **26918** | **2.109%** |
| X-n110-k13 | 14971 | 15223 | 1.683% | 15151 | 1.202% | **15089** | **0.788%** | 15196 | 1.503% | 15407 | 2.912% |
| X-n115-k10 | 12747 | 16113 | 26.406% | 14785 | 15.988% | 13847 | 8.629% | **13325** | **4.534%** | 13513 | 6.009% |
| X-n120-k6 | 13332 | 14085 | 5.648% | 13931 | 4.493% | 14089 | 5.678% | 13833 | 3.758% | **13657** | **2.438%** |
| X-n125-k30 | 55539 | 58513 | 5.355% | 60687 | 9.269% | 58944 | 6.131% | 58603 | 5.517% | **57615** | **3.738%** |
| X-n129-k18 | 28940 | **29246** | **1.057%** | 30332 | 4.810% | 29802 | 2.979% | 29457 | 1.786% | 29309 | 1.275% |
| X-n134-k13 | 10916 | **11302** | **3.536%** | 11581 | 6.092% | 11353 | 4.003% | 11398 | 4.416% | 11363 | 4.095% |
| X-n139-k10 | 13590 | 14035 | 3.274% | 13911 | 2.362% | **13825** | **1.729%** | 13800 | 1.545% | 13911 | 2.362% |
| X-n143-k7 | 15700 | 16131 | 2.745% | 16660 | 6.115% | 16125 | 2.707% | 16147 | 2.847% | **15955** | **1.624%** |
| X-n148-k46 | 43448 | 49328 | 13.533% | 50782 | 16.880% | 46758 | 7.618% | 45599 | 4.951% | **45463** | **4.638%** |
| X-n153-k22 | 21220 | 32476 | 53.040% | 26237 | 23.643% | 23793 | 12.125% | **23316** | **9.877%** | 23340 | 9.991% |
| X-n157-k13 | 16876 | 17660 | 4.646% | 17510 | 3.757% | 17650 | 4.586% | 17410 | 3.164% | **17161** | **1.689%** |
| X-n162-k11 | 14138 | 14889 | 5.312% | 14720 | 4.117% | 14654 | 3.650% | 14662 | 3.706% | **14487** | **2.469%** |
| X-n167-k10 | 20557 | 21822 | 6.154% | 21399 | 4.096% | 21340 | 3.809% | 21275 | 3.493% | **21053** | **2.413%** |
| X-n172-k51 | 45607 | 49556 | 8.659% | 56385 | 23.632% | 51292 | 12.465% | 49073 | 7.600% | **47850** | **4.918%** |
| X-n176-k26 | 47812 | 54197 | 13.354% | 57637 | 20.549% | 55520 | 16.121% | 52727 | 10.280% | **52476** | **9.755%** |
| X-n181-k23 | 25569 | 37311 | 45.923% | 26219 | 2.542% | 26258 | 2.695% | 26241 | 2.628% | **25919** | **1.369%** |
| X-n186-k15 | 24145 | 25222 | 4.461% | 25000 | 3.541% | 25182 | 4.295% | **24836** | **2.862%** | 24711 | 2.344% |
| X-n190-k8 | 16980 | 18315 | 7.862% | 18113 | 6.673% | 18327 | 7.933% | 18113 | 6.673% | **17539** | **3.292%** |
| X-n195-k51 | 44225 | 49158 | 11.154% | 54090 | 22.306% | 49984 | 13.022% | 48185 | 8.954% | **46301** | **4.694%** |
| X-n200-k36 | 58578 | 64618 | 10.311% | 61654 | 5.251% | 61530 | 5.039% | 61483 | 4.959% | **60978** | **4.097%** |
| X-n209-k16 | 30656 | 32212 | 5.076% | 32011 | 4.420% | 32033 | 4.492% | 32055 | 4.564% | **31536** | **2.871%** |
| X-n219-k73 | 117595 | 133545 | 13.564% | 119887 | 1.949% | 121046 | 2.935% | 120421 | 2.403% | **118499** | **0.769%** |
| X-n228-k23 | 25742 | 48689 | 89.142% | 33091 | 28.549% | 31054 | 20.636% | 28561 | 10.951% | **28156** | **9.378%** |
| X-n237-k14 | 27042 | 29893 | 10.543% | 28472 | 5.288% | 28550 | 5.577% | 28486 | 5.340% | **27789** | **2.762%** |
| X-n247-k50 | 37274 | 56167 | 50.687% | 45065 | 20.902% | 43673 | 17.167% | 41800 | 12.143% | **41106** | **10.281%** |
| X-n251-k28 | 38684 | 40263 | 4.082% | 40614 | 4.989% | 41022 | 6.044% | 40822 | 5.527% | **39877** | **3.084%** |
| Avg. Gap | | | 16.629% | | 9.820% | | 6.884% | | 5.160% | | **4.025%** |

# E   Licenses for Code and Datasets

The licenses for the codes and the datasets used in this work are listed in Table 9.

# F   Broader Impacts

This research contributes to the field of neural combinatorial optimization by employing advanced machine learning techniques to address large-scale and multi-variant VRP problems. We believe that the proposed multi-task knowledge distillation training framework, heavy decoder model, and Random Reordering Reconstruction (R3C) strategy can provide valuable insights and inspire subsequent work to explore more efficient and effective neural methods for solving large-scale and diverse VRP problems. As a general learning-based approach to solving the VRP, the proposed multi-task knowledge distillation training framework and R3C strategy do not inherently possess any specific potential negative social impacts.

Table 7: Results on VRPTW instances (Set-Solomon) [39]. All models are trained on instances of size $N = 100$, and inference utilized data augmentation [11] and greedy rollout by default.

| Set-Solomon | | POMO | | POMO-MTL | | MVMoE/4E | | MVMOE/4E-L | | MTL-KD | |
|---|---|---|---|---|---|---|---|---|---|---|---|---|
| Instance | Opt. | Obj. | Gap | Obj. | Gap | Obj. | Gap | Obj. | Gap | Obj. | Gap |
| R101 | 1637.7 | 1805.6 | 10.252% | 1821.2 | 11.205% | 1798.1 | 9.794% | **1730.1** | **5.641%** | 1768.0 | 7.956% |
| R102 | 1466.6 | **1556.7** | **6.143%** | 1596.0 | 8.823% | 1572.0 | 7.187% | 1574.3 | 7.345% | 1564.4 | 6.668% |
| R103 | 1208.7 | 1341.4 | 10.979% | **1327.3** | **9.812%** | 1328.2 | 9.887% | 1359.4 | 12.470% | 1379.0 | 14.090% |
| R104 | 971.5 | 1118.6 | 15.142% | 1120.7 | 15.358% | 1124.8 | 15.780% | 1098.8 | 13.100% | **1074.1** | **10.561%** |
| R105 | 1355.3 | 1506.4 | 11.149% | 1514.6 | 11.754% | 1479.4 | 9.157% | **1456.0** | **7.433%** | 1465.2 | 8.109% |
| R106 | 1234.6 | 1365.2 | 10.578% | 1380.5 | 11.818% | 1362.4 | 10.352% | 1353.5 | 9.627% | **1346.1** | **9.031%** |
| R107 | 1064.6 | 1214.2 | 14.052% | 1209.3 | 13.592% | **1182.1** | **11.037%** | 1196.5 | 12.391% | 1193.9 | 12.145% |
| R108 | 932.1 | 1058.9 | 13.604% | 1061.8 | 13.915% | **1023.2** | **9.774%** | 1039.1 | 11.481% | 1035.4 | 11.082% |
| R109 | 1146.9 | 1249.0 | 8.902% | 1265.7 | 10.358% | 1255.6 | 9.478% | **1224.3** | **6.750%** | 1260.0 | 9.861% |
| R110 | 1068.0 | 1180.4 | 10.524% | 1171.4 | 9.682% | 1185.7 | 11.021% | **1160.2** | **8.635%** | 1199.5 | 12.313% |
| R111 | 1048.7 | 1177.2 | 12.253% | 1211.5 | 15.524% | **1176.1** | **12.148%** | 1197.8 | 14.220% | 1178.7 | 12.396% |
| R112 | 948.6 | 1063.1 | 12.070% | 1057.0 | 11.427% | 1045.2 | 10.183% | **1044.2** | **10.082%** | 1057.2 | 11.448% |
| RC101 | 1619.8 | 2643.0 | 63.168% | 1833.3 | 13.181% | 1774.4 | 9.544% | **1749.2** | **7.988%** | 1774.0 | 9.520% |
| RC102 | 1457.4 | **1534.8** | **5.311%** | 1546.1 | 6.086% | 1544.5 | 5.976% | 1556.1 | 6.771% | 1588.4 | 8.989% |
| RC103 | 1258.0 | 1407.5 | 11.884% | **1396.2** | **10.986%** | 1402.5 | 11.486% | 1415.3 | 12.502% | 1451.8 | 15.405% |
| RC104 | 1132.3 | 1261.8 | 11.437% | 1271.7 | 12.311% | 1265.4 | 11.755% | 1264.2 | 11.649% | **1241.6** | **9.653%** |
| RC105 | 1513.7 | **1612.9** | **6.553%** | 1644.9 | 8.668% | 1635.5 | 8.047% | 1619.4 | 6.980% | 1688.3 | 11.535% |
| RC106 | 1372.7 | 1539.3 | 12.137% | 1552.8 | 13.120% | 1505.0 | 9.638% | 1509.5 | 9.968% | **1468.8** | **7.001%** |
| RC107 | 1207.8 | 1347.7 | 11.583% | 1384.8 | 14.655% | 1351.6 | 11.906% | **1324.1** | **9.625%** | 1367.6 | 13.231% |
| RC108 | 1114.2 | 1305.5 | 17.169% | 1274.4 | 14.378% | 1254.2 | 12.565% | 1247.2 | 11.939% | **1232.3** | **10.600%** |
| RC201 | 1261.8 | 2045.6 | 62.118% | 1761.1 | 39.570% | 1577.3 | 25.004% | 1517.8 | 20.285% | **1490.8** | **18.149%** |
| RC202 | 1092.3 | 1805.1 | 65.257% | 1486.2 | 36.062% | 1616.5 | 47.990% | 1480.3 | 35.520% | **1336.5** | **22.356%** |
| RC203 | 923.7 | 1470.4 | 59.186% | 1360.4 | 47.277% | 1473.5 | 59.521% | 1479.6 | 60.182% | **1237.8** | **34.005%** |
| RC204 | 783.5 | 1323.9 | 68.973% | 1331.7 | 69.968% | 1286.6 | 64.212% | 1232.8 | 57.342% | **1066.0** | **36.056%** |
| RC205 | 1154.0 | 1568.4 | 35.910% | 1539.2 | 33.380% | 1537.7 | 33.250% | **1440.8** | **24.850%** | 1448.3 | 25.503% |
| RC206 | 1051.1 | 1707.5 | 62.449% | 1472.6 | 40.101% | 1468.9 | 39.749% | 1394.5 | 32.671% | **1354.7** | **28.884%** |
| RC207 | 962.9 | 1567.2 | 62.758% | 1375.7 | 42.870% | 1442.0 | 49.756% | 1346.4 | 39.831% | **1235.9** | **28.352%** |
| RC208 | 776.1 | 1505.4 | 93.970% | 1185.6 | 52.764% | 1107.4 | 42.688% | 1167.5 | 50.437% | **1064.0** | **37.096%** |
| Avg. Gap | | 28.054% | | 21.380% | | 20.317% | | 18.490% | | **15.786%** | |

Table 8: Results on large-scale CVRPLIB instances (Set-X) [38]. All models are trained on instances of size $N = 100$, and inference utilizes data augmentation [11] and greedy rollout by default.

| Set-X Instance | Opt. | POMO Obj. | POMO Gap | LEHD Obj. | LEHD Gap | POMO-MTL Obj. | POMO-MTL Gap | MVMoE/4E Obj. | MVMoE/4E Gap | MVMOE/4E-L Obj. | MVMOE/4E-L Gap | RF-MVMOE Obj. | RF-MVMOE Gap | RF-TE Obj. | RF-TE Gap | MTL-KD Obj. | MTL-KD Gap |
|---|---|---|---|---|---|---|---|---|---|---|---|---|---|---|---|---|---|
| X-n502-k39 | 69226 | 75617 | 9.232% | 71438 | 3.195% | 77284 | 11.640% | 73533 | 6.222% | 74429 | 7.516% | 76338 | 10.274% | 71791 | 3.705% | **71124** | **2.742%** |
| X-n513-k21 | 24201 | 30518 | 26.102% | **25624** | **5.880%** | 28510 | 17.805% | 32102 | 32.647% | 31231 | 29.048% | 32639 | 34.866% | 28465 | 17.619% | 25947 | 7.215% |
| X-n524-k153 | 154593 | 201877 | 30.586% | 280556 | 81.480% | 192249 | 24.358% | 186540 | 20.665% | 182392 | 17.982% | **170999** | **10.612%** | 174381 | 12.800% | 171306 | 10.811% |
| X-n536-k96 | 94846 | 106073 | 11.837% | 103785 | 9.425% | 106514 | 12.302% | 109581 | 15.536% | 108543 | 14.441% | 105847 | 11.599% | 103272 | 8.884% | **101893** | **7.430%** |
| X-n548-k50 | 86700 | 103093 | 18.908% | 90644 | 4.549% | 94562 | 9.068% | 95894 | 10.604% | 95917 | 10.631% | 104289 | 20.287% | 100956 | 16.443% | **89169** | **2.848%** |
| X-n561-k42 | 42717 | 49370 | 15.575% | **44728** | **4.708%** | 47846 | 12.007% | 56008 | 31.114% | 51810 | 21.287% | 53383 | 24.969% | 49454 | 15.771% | 45467 | 6.438% |
| X-n573-k30 | 50673 | 83545 | 64.871% | 53482 | 5.543% | 60913 | 20.208% | 59473 | 17.366% | 57042 | 12.569% | 61524 | 21.414% | 55952 | 10.418% | **53466** | **5.512%** |
| X-n586-k159 | 190316 | 229887 | 20.792% | 232867 | 22.358% | 208893 | 9.761% | 215668 | 13.321% | 214577 | 12.748% | 212151 | 11.473% | 205575 | 8.018% | **200863** | **5.542%** |
| X-n599-k92 | 108451 | 150572 | 38.839% | 115377 | 6.386% | 120333 | 10.956% | 128949 | 18.901% | 125279 | 15.517% | 126578 | 16.714% | 116560 | 7.477% | **113513** | **4.668%** |
| X-n613-k62 | 59535 | 68451 | 14.976% | **62484** | **4.953%** | 67984 | 14.192% | 82586 | 38.718% | 74945 | 25.884% | 73456 | 23.383% | 67267 | 12.987% | 63035 | 5.879% |
| X-n627-k43 | 62164 | 84434 | 35.825% | 67568 | 8.693% | 73060 | 17.528% | 70987 | 14.193% | 70905 | 14.061% | 70414 | 13.271% | 67572 | 8.700% | **65755** | **5.777%** |
| X-n641-k35 | 63682 | 75573 | 18.672% | 68249 | 7.172% | 72643 | 14.071% | 75329 | 18.289% | 72655 | 14.090% | 71975 | 13.023% | 70831 | 11.226% | **67593** | **6.141%** |
| X-n655-k131 | 106780 | 127211 | 19.134% | 117532 | 10.069% | 116988 | 9.560% | 117678 | 10.206% | 118475 | 10.952% | 119057 | 11.497% | 112202 | 5.078% | **109748** | **2.780%** |
| X-n670-k130 | 146332 | 208079 | 42.197% | 220927 | 50.977% | 190118 | 29.922% | 197695 | 35.100% | 183447 | 25.364% | 168226 | 14.962% | 168999 | 15.490% | **161076** | **10.076%** |
| X-n685-k75 | 68205 | 79482 | 16.534% | **72946** | **6.951%** | 80892 | 18.601% | 97388 | 42.787% | 89441 | 31.136% | 82269 | 20.620% | 77847 | 14.137% | 73249 | 7.395% |
| X-n701-k44 | 81923 | 97843 | 19.433% | 86327 | 5.376% | 92075 | 12.392% | 98469 | 20.197% | 94924 | 15.870% | 90189 | 10.090% | 89932 | 9.776% | **85967** | **4.936%** |
| X-n716-k35 | 43373 | 51381 | 18.463% | **46502** | **7.214%** | 52709 | 21.525% | 56773 | 30.895% | 52305 | 20.593% | 52250 | 20.467% | 49669 | 14.516% | 47012 | 8.390% |
| X-n733-k159 | 136187 | 159098 | 16.823% | 149115 | 9.493% | 161961 | 18.925% | 178322 | 30.939% | 167477 | 22.976% | 156387 | 14.833% | 148463 | 9.014% | **142712** | **4.791%** |
| X-n749-k98 | 77269 | 87786 | 13.611% | 83439 | 7.985% | 90582 | 17.229% | 100438 | 29.985% | 94497 | 22.296% | 92147 | 19.255% | 85171 | 10.227% | **82295** | **6.505%** |
| X-n766-k71 | 114417 | 135464 | 18.395% | 131487 | 14.919% | 144041 | 25.891% | 152352 | 33.155% | 136255 | 19.086% | 130505 | 14.061% | 129935 | 13.563% | **123310** | **7.772%** |
| X-n783-k48 | 72386 | 90289 | 24.733% | **76766** | **6.051%** | 83169 | 14.897% | 100383 | 38.677% | 92960 | 28.423% | 96336 | 33.087% | 83185 | 14.919% | 77332 | 6.833% |
| X-n801-k40 | 73305 | 124278 | 69.536% | **77546** | **5.785%** | 85077 | 16.059% | 91560 | 24.903% | 87662 | 19.585% | 87118 | 18.843% | 86164 | 17.542% | 78041 | 6.460% |
| X-n819-k171 | 158121 | 193451 | 22.344% | 178558 | 12.925% | 177157 | 12.039% | 183599 | 16.113% | 185832 | 17.525% | 179596 | 13.581% | 174441 | 10.321% | **170672** | **7.938%** |
| X-n837-k142 | 193737 | 237884 | 22.787% | 207709 | 7.212% | 214207 | 10.566% | 229526 | 18.473% | 221286 | 14.220% | 230362 | 18.904% | 208528 | 7.635% | **203165** | **4.866%** |
| X-n856-k95 | 88965 | 152528 | 71.447% | **92936** | **4.464%** | 101774 | 14.398% | 99129 | 11.425% | 106816 | 20.065% | 105801 | 18.924% | 98291 | 10.483% | 94079 | 5.748% |
| X-n876-k59 | 99299 | 119764 | 20.609% | **104183** | **4.918%** | 116617 | 17.440% | 119619 | 20.463% | 114333 | 15.140% | 114016 | 14.821% | 107416 | 8.174% | 105873 | 6.620% |
| X-n895-k37 | 53860 | 70245 | 30.421% | **58028** | **7.739%** | 65587 | 21.773% | 79018 | 46.710% | 64310 | 19.402% | 69099 | 28.294% | 64871 | 20.444% | 58606 | 8.812% |
| X-n916-k207 | 329179 | 399372 | 21.324% | 385208 | 17.021% | 361719 | 9.885% | 383681 | 16.557% | 374016 | 13.621% | 373600 | 13.494% | 352998 | 7.236% | **346940** | **5.396%** |
| X-n936-k151 | 132715 | 237625 | 79.049% | 196547 | 48.097% | 186262 | 40.347% | 220926 | 66.466% | 190407 | 43.471% | 161343 | 21.571% | 163162 | 22.942% | **152094** | **14.602%** |
| X-n957-k87 | 85465 | 130850 | 53.104% | **90295** | **5.651%** | 98198 | 14.898% | 113882 | 33.250% | 105629 | 23.593% | 123633 | 44.659% | 102689 | 20.153% | 90407 | 5.782% |
| X-n979-k58 | 118976 | 147687 | 24.132% | 127972 | 7.561% | 138092 | 16.067% | 146347 | 23.005% | 139682 | 17.404% | 131754 | 10.740% | 129952 | 9.225% | **127650** | **7.291%** |
| X-n1001-k43 | 72355 | 100399 | 38.759% | **76689** | **5.990%** | 87660 | 21.153% | 114448 | 58.176% | 94734 | 30.929% | 88969 | 22.962% | 85929 | 18.760% | 78833 | 8.953% |
| Avg. Gap | | | 29.658% | | 12.836% | | 16.796% | | 26.408% | | 19.607% | | 18.795% | | 12.303% | | **6.655%** |

Table 9: Licenses of Used Codes and Datasets

| Type | Resource | License Type | URL / Reference |
|---|---|---|---|
| Code | MVMoE [24] | MIT License | https://github.com/RoyalSkye/Routing-MVMoE |
| Code | MT-POMO [23] | MIT License | https://github.com/FeiLiu36/MTNCO |
| Code | HGS-PyVRP [36] | MIT License | https://github.com/PyVRP/PyVRP |
| Code | OR-Tools [9] | Apache License | https://github.com/google/or-tools |
| Dataset | CVRPLIB(Set-X) [38] | Available for academic research use | http://vrp.galgos.inf.puc-rio.br/index.php/en/ |
| Dataset | Solomon [39] | Available for academic research use | https://www.sintef.no/projectweb/top/vrptw/solomon-benchmark/ |

