# OpenReview forum: "MTL-KD: Multi-Task Learning Via Knowledge Distillation for Generalizable Neural Vehicle Routing Solver"
_NeurIPS.cc/2025/Conference — NeurIPS 2025 poster_

### Official Review · Reviewer_zdRP · 2025-06-30

**Clarity:** 2
**Significance:** 3
**Originality:** 3
**Rating:** 3
**Confidence:** 3

**Summary:**

The paper presents an extension of Neural Combinatorial Optimization (NCO).

The proposed approach considers smaller instances and multiple smaller and easier models trained on the smaller problems.

The smaller models are trained first (pre-trained models).

Then the final model is trained in a way similar to multi-task learning, where the largest model has an extended loss function that includes all the instances and the associated small pre-trained models, at the same time.

**Questions:**

The smaller models represent different types of problems; for example, VRP with or without Time windows.

Why should the method take advantage of the instances of other types of problems?

**Ethical Concerns:**

["NO or VERY MINOR ethics concerns only"]

**Final Justification:**

I have not changed my score. In my initial review, I did not fully understand the context.

The paper has two contributions: 1) the use of knowledge distillation to train the multi-task LDHE, 2) the use of data augmentation with a variant of the partial solutions reconstruction.

The paper then shows improved results with the state-of-the-art models.

On one side, I think the paper has a proper contribution; on the other side, the presentation of the information is confusing and not self-contained.

**Limitations:**

The approach does not seem to have any criticality, so there does not seem to be a necessity for a limitation session.

The authors describe the limitations of the work in the last session.

**Paper Formatting Concerns:**

I have not found any

**Quality:**

2

**Strengths And Weaknesses:**

Strengths :
1. From the experimental session, it is possible to see that the method produces better solutions
2. The paper describes the protocol for training that is relatively intuitive
3. relatively extensive experiments and comparison with other baselines

Weaknesses
1. The method uses many known results (MTL, NCO, knowledge distillation), and does not appear to be very innovative

---

> ### Author Rebuttal · Authors · 2025-07-31
>
> Dear Reviewer zdRP,
>
> Thank you very much for your thorough review and valuable feedback on our work. We have carefully considered your comments and would like to provide detailed elaborations and clarifications for the points you raised.
>
> ---
>
> ### Weakness 1: The method uses many known results (MTL, NCO, knowledge distillation), and does not appear to be very innovative.
>
> Thank you very much for raising this concern. We fully understand the reviewer's concern regarding innovation. As we elaborate below, the innovation of our work does not lie in inventing isolated techniques, but rather in **synthesizing them through a novel architecture and strategy** to effectively solve a series of critical challenges, from model training to inference.
> Specifically:
>
> + Firstly, concerning the **training paradigm**, our core innovation validates that in the NCO domain, **knowledge distillation (KD) can be leveraged to allow multiple easy-to-train single-task "teacher" models to provide label-free guidance to a unified, heavy-decoder multi-task "student" model.** Previously, heavy decoders were difficult to apply efficiently in multi-task scenarios due to their reliance on training labels. Our design **is the first to enable this technical path**, and its architectural innovation is a core value of our work.
>
> + Secondly, regarding the **inference strategy**, our proposed **R3C (Random Reordering Re-Construct) is entirely new**. Existing reconstruction methods (e.g., RRC) primarily rely on "flipping" sub-paths to increase diversity. However, this has limited effectiveness and can even disrupt the feasibility of certain VRP tasks (e.g., VRPTW). The core innovation of R3C lies in **randomly reordering the external sequence of various sub-paths**. This creates far more diverse sub-problems at a structural level than "flipping," more effectively helping the model escape local optima and further improving performance.
>
> ---
>
> ### Question 1: The smaller models represent different types of problems; for example, VRP with or without Time windows. Why should the method take advantage of the instances of other types of problems?
>
> Thank you very much for raising this crucial question about leveraging cross-task data. The motivation behind our work to build a unified multi-task model is precisely to address the fundamental limitations of single-task methods in generalization ability and real-world applications.
>
> 1.  **Zero-Shot Generalization:** By co-training on diverse VRP variants, the model is compelled to learn a more **generic problem representation** rather than "overfitting" to specific constraints. This understanding of the problem's inherent structure enables it to perform **zero-shot generalization** to new tasks (i.e., new combinations of constraints) not encountered during training. This is crucial for tackling the ever-emerging challenges in the real world.
> 2.  **Practicality and Scalability:** From a practical perspective, real-world routing problems are often complex combinations of various constraints. Maintaining a single-task model for every possible combination would lead to a "combinatorial explosion," making deployment and maintenance impractical. A **unified model significantly simplifies engineering processes**, offering a more flexible and scalable solution, making its widespread application in industry possible.
>
> ---
>
> Once again, we would like to express our heartfelt gratitude for the time and effort you've dedicated to reviewing our work. We sincerely hope that our response can address your concerns.

---

> ### Comment · Reviewer_zdRP · 2025-08-04
> **Reply to authors**
>
> Dear Authors,
>
> I would like to thank you for the clarification.
>
> > Zero-Shot Generalization
>
> > Practicality and Scalability
>
> These are good arguments. Where do you observe them?
>
> >  knowledge distillation (KD) can be leveraged to allow multiple easy-to-train single-task "teacher" models to provide label-free guidance to a unified, heavy-decoder
>
> > R3C (Random Reordering Re-Construct) is entirely new.
>
> Both the previous points seem to be heavily based on LEHD.
> Is it correct that you do not compare with the original LEHD? why it is so?
> You have a session, "KD vs. RL" but both build on LEHD. Is that correct?
>
> Looking forward to the additional clarification.

---

> ### Author Response · Authors · 2025-08-05
> **Response to Reviewer zdRP (1/2)**
>
> Dear Reviewer zdRP,
>
> Thank you very for the follow-up questions and for giving us the opportunity to clarify. We would like to make the following clarifications to address your questions.
>
>
> #### Q1: Where do you observe the model's zero-shot generalization, practicality, and scalability?
>
> In our experimental design, our unified model is trained on only 6 fundamental VRP variants of problem size $n$=100, and then is tested on a total number of 16 VRP variants of problem sizes $n$=100, 500, and 1000, with 6 seen and 10 unseen VRP variants. We compare our model with two representative baseline neural solvers, MT-POMO and MVMoE, on these unseen VRP problems with varying problem sizes from 100 to 1k. The results are shown in **Tables 1 and 2 in the main paper**. **The above arguments**, i.e., zero-shot generalization practicality, and scalability of our method **are observed from these tables.** For the ease of reading, we present a part of the results in the table below, which shows the results on 10 unseen VRP variants of varying problem sizes. We make the following clarification to observe the arguments.
>
>
> - **Zero-Shot Generalization:** From these results, we can observe that without any fine-tuning or additional training, our model achieves the lowest gaps on 9 out of 10 unseen VRPs of problem size $n$=100, and achieves the lowest gaps on all 10 unseen VRPs of larger problem size $n$=500 and $n$=1000. These results demonstrate the promising zero-shot generalization ability of our model.
>
> - **Practicality:** Our single unified model can handle a total number of 16 VRP variants, with 6 seen and 10 unseen. This effectively avoids the "combinatorial explosion" problem of training and maintaining a specialized model for every possible combination of constraints, which effectively reduces a substantial amount of training budget.
>
> - **Scalability:** Our model, only trained in problems with size $n$=100, can effectively scale to different problem sizes $n$=500 and 1000, achieving the lowest gaps in most cases. This highlights the good scalability of our model.
>
>
>
> Table 1: Performance in terms of gap of MTL-KD and baseline neural solvers on Ten Unseen Tasks. Gap measures the difference between solutions achieved by different methods and the solutions obtained using the classical solvers HGS or OR-Tools. The full table can be found in Table 2 in the main paper.
> | Method | Problem | Problem Size |  |  | Problem | Problem Size |  |  |
> |---|:---:|:---:|:---:|:---:|:---:|:---:|:---:|:---:|
> |  | OVRPL | n=100 | n=500 | n=1k | VRPLTW | n=100 | n=500 | n=1k |
> | MT-POMO(M+aug8) |  | 4.72\% | 18.95\% | 29.08\% |  | 5.21\% | 26.95\% | 34.99\% |
> | MVMoE(M+aug8) |  | 4.43\% | 31.36\% | 78.48\% |  | 4.84\% | 28.45\% | 48.79\% |
> | MTL-KD(R3C200)$ _{128}$ |  |  **4.29\%** |  **7.41\%**  | **9.12\%**|  | **4.54\%** | **6.51\%** | **8.14\%** |
> |  | OVRPLTW | n=100 | n=500 | n=1k | OVRPB | n=100 | n=500 | n=1k |
> | MT-POMO(M+aug8) |  | 6.39\% | 38.95\% | 63.87\% |  | 7.34\% | 9.28\% | 21.95\% |
> | MVMoE(M+aug8) |  | 5.90\% | 58.68\% | 136.41\% |  | 7.10\% | 34.77\% | 98.81\% |
> | MTL-KD(R3C200)$ _{128}$ |  | **5.02\%** | **8.34\%** | **12.22\%** |  | **4.95\%** | **-4.16\%**  | **-4.48\%** |
> |  | VRPBL | n=100 | n=500 | n=1k | VRPBTW | n=100 | n=500 | n=1k |
> | MT-POMO(M+aug8) |  | 0.66\% | 0.45\% | 4.94\% |  | 6.04\% | 20.89\% |29.22\% |
> | MVMoE(M+aug8) |  |**0.28\%** | 32.63\% |62.10\% |  | 5.82\% | 22.16\% | 41.85\% |
> | MTL-KD(R3C200)$ _{128}$ |  | 0.55\% | **-7.64\%** | **-8.08\%**  |  |**4.84\%** |**2.53\%** | **5.37\%** |
> |  | OVRPBL | n=100 | n=500 | n=1k | OVRPBTW | n=100 | n=500 | n=1k |
> | MT-POMO(M+aug8) |  | 7.35\% | 10.13\% | 23.13\% |  | 10.39\% | 30.96\% | 54.87\% |
> | MVMoE(M+aug8) |  | 7.12\% | 36.29\% | 99.86\% |  | 9.90\% | 43.46\% | 112.77\% |
> | MTL-KD(R3C200)$ _{128}$ |  | **5.95\%** | **-1.26\%** |  **-0.98\%** |  | **8.30\%** | **3.62\%**  | **7.17\%** |
> |  | VRPBLTW | n=100 | n=500 | n=1k | OVRPBLTW | n=100 | n=500 | n=1k |
> | MT-POMO(M+aug8) |  | 7.52\% | 19.91\% | 29.74\% |  | 10.45\% | 30.01\% | 53.28\% |
> | MVMoE(M+aug8) |  | 7.10\% |21.43\% | 42.16\% |  | 9.97\% | 42.90\% | 112.07\% |
> | MTL-KD(R3C200)$ _{128}$ |  | **7.03\%**  |  **3.26\%** | **6.33\%** |  | **9.84\%** |  **4.41\%** | **8.11\%** |

---

> ### Author Response · Authors · 2025-08-05
> **Response to Reviewer zdRP (2/2)**
>
> #### Q2: Both the previous points seem to be heavily based on LEHD. Is it correct that you do not compare with the original LEHD? why it is so? You have a session, "KD vs. RL" but both build on LEHD. Is that correct?
>
>
> Thank you very much for raising these questions. We would like to make the following clarification to address your questions point-by-point.
>
> **1. The Base Model**
>
> We acknowledge that the two points mentioned are heavily based on LEHD. We select the LEHD model as the base model since it demonstrates the promising generalization ability. However, due to its reliance on supervised learning (SL)for training, it is impractical to obtain a substantial amount (e.g., one million) of near-optimal instance-solution pairs for SL to train the LEHD model on each VRP variant. It motivates us to propose the label-free MTL-KD (multi-task learning method via knowledge distillation) method to tackle this issue. Moreover, we propose R3C, which is specifically designed for the 16 VRP tasks and further boosts the performance of the multi-task model.
>
> **2. Comparison with the Original LEHD Model**
>
> We did not directly compare our model with the original single-task LEHD model on the 16 VRP variants in the main text. The main reason is that the original LEHD model is only applicable to TSP or CVRP problems, and as mentioned before, it is computationally impractical to train the LEHD model on each VRP variant. Therefore, a comprehensive comparison across all 16 VRP problems may not be possible.
>
>
> Nonetheless, on the CVRP problem, we did include a comparison on a real-world dataset (CVRPLIB Set-X), with results shown in **Appendix C, Table 7**. This result shows that our model's gap to the baseline solution is only 6.655%, which is superior to the single-task LEHD model's 12.836% gap. This highlights our model's stronger generalization ability in real-world scenarios.
>
> Moreover, we would like to clarify that in this paper, our primary research objective is to design a generalizable neural vehicle routing solver for the multi-task domain. Therefore, we mainly compare our MTL-KD model with the recent advanced multi-task models such as MVMoE.
>
>
> **3. Explanation of the "KD vs. RL" Experiment**
>
> We sincerely apologize for the misleading description "we compared the MTL-KD model with a LEHD model trained using RL" in line 217 in the subsection 'KD vs. RL'. We clarify that in this section, both models are built on the same multi-task LEHD model architecture proposed in our paper.
>
> In this subsection 'KD vs. RL', the experiment's purpose is to compare two different training paradigms on the exact same multi-task LEHD model architecture: **knowledge distillation (KD) and traditional reinforcement learning (RL)**. The results in **Table 3 in the main paper** clearly show that for heavy-decoder models, the distillation-based training paradigm of MTL-KD is far more effective than the RL method in efficiently training the multi-task LEHD model and leads to superior performance.
>
> ---
>
> Once again, we sincerely thank you for your valuable questions and hope that our responses have adequately addressed your concerns.

---

> > ### Comment · Reviewer_zdRP · 2025-08-07
> >
> > Dear authors,
> >
> > > Zero-Shot
> >
> > I see that this naming was introduced in MVMoE paper. While I understood after reading that you were testing on different problems, it was not clear how this was happening, only after reading the annex, this become clear. I find this not helpful.
> >
> > > MT and KD
> >
> > It seems that the MT problem was already introduced in MVMoE. Exploring KD is a proper contribution.
> >
> > > LEHD
> >
> > The comparison with LEHD (or the lack thereof) and the justification (SL vs RL), as you explained now, should be clearer in the presentation.
> >
> > I thank the authors for the clarification. I would prefer to leave the score as it is and provide feedback during the following phase.

---

> > > ### Author Response · Authors · 2025-08-08
> > >
> > > Dear Reviewer zdRP,
> > >
> > > Thank you very much for your valuable feedback. We would like to take this opportunity to provide further clarification.
> > >
> > > **1. On Zero-Shot**
> > >
> > > Zero-shot generalization is an important concept in machine learning that describes a model's capability to solve unseen tasks without requiring training on them. This has been widely studied in fields such as computer vision, natural language processing (e.g., [1-3]), and also neural combinatorial optimization (NCO). **It is not originally proposed by the MVMoE paper**. In the context of NCO, the zero-shot paradigm refers to a single, unified model, trained on several specific VRP problems, that can solve unseen VRP problems [4-5] without additional training. In our experiments, our model is trained on 6 seen tasks and demonstrates its zero-shot generalization ability on 10 unseen tasks. Thanks to our carefully designed model structure, it can handle a unified input representation of constraints like capacity, time windows, distance, and backhauls, allowing it to effectively solve unseen different VRP variants during the test phase.
> > >
> > >
> > > **2. On Multi-Task Learning and Our Contributions**
> > >
> > > Thank you very much for noting that multi-task learning has been explored in prior work. We would like to clarify that **multi-task learning** is not a single innovation proposed by us or MVMoE, but rather an emerging **research subfield** within NCO [4-5]. The primary goal of this subfield is to develop generic solvers that can handle multiple VRP variants using a unified model.
> > >
> > > Within this context, our work makes the following concrete and substantial contributions to the field of multi-task learning:
> > >
> > > * **Introducing an Innovative Training Paradigm for Multi-task VRP:** The original heavy decoder model demonstrates promising scalability, i.e., only trained on small-scale instances (e.g., TSP100), the model can generate high-quality solutions on large-scale instances (e.g., TSP1000). But it needs a substantial amount of labeled data for model training via supervised learning. We address this challenge by proposing an innovative label-free **multi-task knowledge distillation** method. This approach successfully introduces the powerful heavy decoder model to multi-task VRP solving for the first time, demonstrating its great scalability on each VRP variant in this domain.
> > > * **Designing the R3C Method:** We design the Random Reordering Re-Construct (R3C) method to perform random reconstruction for multiple VRP variants, effectively enhancing the model's performance.
> > > * **Promising performance:** Extensive experiments demonstrate the superior performance of our model on both seen and unseen VRP variants of 100 to 1000 nodes compared to current state-of-the-art multi-task VRP neural solvers.
> > >
> > >
> > >
> > > **3. On the Clarification of the LEHD Model**
> > >
> > > We sincerely apologize again for the clarity in our paper's descriptions. Following your suggestion, we will make the corresponding corrections in the final version.
> > >
> > > Thank you very much again for your detailed and constructive comments. We sincerely wish that our response could address your concerns.
> > >
> > >
> > > [1] Evaluating knowledge transfer and zero-shot learning in a large-scale setting. CVPR 2011.
> > >
> > > [2] Zero-shot learning with semantic output codes. NeurIPS 2009.
> > >
> > > [3] Zero-shot learning via semantic similarity embedding. ICCV 2015.
> > >
> > > [4] Multi-task learning for routing problem with cross-problem zero-shot generalization. KDD 2024.
> > >
> > > [5] Mvmoe: multi-task vehicle routing solver with mixture-of-experts. ICML 2024.

---

### Official Review · Reviewer_sUQS · 2025-07-01

**Clarity:** 3
**Significance:** 3
**Originality:** 3
**Rating:** 4
**Confidence:** 3

**Summary:**

This paper proposes a knowledge distillation-based multi-task VRP solver (MTL-KD) aimed at addressing the generalization limitations of existing methods when handling large-scale VRP problems. MTL-KD employs a lightweight encoder–heavy decoder (LEHD) architecture to train multiple teacher models and transfers their knowledge to a decoder model through knowledge distillation, enabling efficient label-free training and improved generalization. In addition, the paper introduces a Random Reordering Reconstruction (R3C) strategy to further enhance model performance.

**Questions:**

1.	The training process of the student model, particularly how it distills and incorporates knowledge from the teacher models, is not clearly described. The current explanation may leave readers uncertain about the effectiveness and implementation details of the knowledge distillation mechanism. Try to make the training procedure of the student model more explicit in both the textual description and the framework figure.
2.	The encoder used in the current model may not fully capture the complex spatial and relational structures present in large-scale VRP instances. This may limit the model’s expressiveness and scalability. Have the authors considered employing graph-based encoder architectures (e.g., GNNs) to better model the underlying structure of the problem? A discussion or experiment in this direction could enhance the technical depth and applicability of the approach.
3.	Some figures, especially in the ablation study section, contain text that is too small to read clearly. This affects the readability and interpretability of the experimental results. Improve the visual clarity of these figures would make the paper more accessible and reader-friendly.

**Ethical Concerns:**

["NO or VERY MINOR ethics concerns only"]

**Limitations:**

The paper does not explicitly discuss the computational overhead introduced by simultaneously training multiple teacher models alongside the student model. This process is likely to incur significant training costs, which may limit the practicality of the approach in resource-constrained settings. It would be helpful for the authors to provide a discussion on whether this overhead is acceptable. Including such analysis would improve the transparency and practical relevance of the method.

**Quality:**

3

**Strengths And Weaknesses:**

Strengths
•	Quality: The paper is technically sound and addresses an interesting and practical problem. It introduces a knowledge distillation-based approach to tackle the challenges of training models under multi-variant conditions and limited data availability. Experimental results demonstrate that the proposed MTL-KD model achieves strong performance on training tasks, unseen tasks, and real-world datasets, validating its effectiveness and practicality. Moreover, the authors conduct extensive comparative and ablation studies to verify the model's design.
•	Clarity: The paper is well-structured, and the writing is fluent. The experimental results and conclusions are clearly articulated. Detailed descriptions of the model architecture and experimental setup help readers understand the main contributions. In addition, the use of well-designed diagrams enhances the clarity of the proposed architecture and the applied optimizations.
•	Significance: The findings of this study are significant for both the VRP and Neural Combinatorial Optimization (NCO) communities. The proposed MTL-KD model demonstrates advantages in both performance and generalization, offering a novel approach to solving large-scale VRP problems.
•	Originality: The paper applies knowledge distillation techniques to the VRP domain and introduces a novel Random Reordering Reconstruction (R3C) strategy, showcasing a degree of innovation.
Weaknesses
•	The encoder architecture used in the current model appears relatively simplistic for capturing the intricate relational structures present in large-scale VRP problems. This might hinder the model’s overall representational capacity and performance.
•	The description of the training process for the student model is somewhat unclear, which may lead to confusion regarding its training effectiveness. Additionally, the text in the ablation study figures is too small and difficult to read.
Suggestions
•	Consider exploring more powerful encoder designs, such as graph neural network-based encoders, to improve scalability and performance.
•	Improve the clarity of the figures, particularly in the ablation study section, and provide a clearer explanation of the student model training process.
Overall, this paper demonstrates notable strengths in terms of quality, clarity, significance, and originality. However, there are still some areas that require improvement.

---

> ### Author Rebuttal · Authors · 2025-07-31
>
> Dear Reviewer sUQS,
>
> Thank you for your insightful comments and questions regarding our paper. We truly appreciate your thorough review and value your feedback, which has helped us enhance our presentation and technical depth. We address your points in detail below.
>
> ---
>
> ### Weakness 1 & Question 2: Encoder Architecture and Scalability
> Thank you very much for raising these concerns. We would like to make the following clarification to address your concerns.
>
> **W1:** The encoder architecture used in the current model appears relatively simplistic for capturing the intricate relational structures present in large-scale VRP problems. This might hinder the model’s overall representational capacity and performance. Consider exploring more powerful encoder designs, such as graph neural network-based encoders, to improve scalability and performance.
>
> **Q2:** The encoder used in the current model may not fully capture the complex spatial and relational structures present in large-scale VRP instances. This may limit the model’s expressiveness and scalability. Have the authors considered employing graph-based encoder architectures (e.g., GNNs) to better model the underlying structure of the problem? A discussion or experiment in this direction could enhance the technical depth and applicability of the approach.
>
> Thank you for raising this excellent point, which touches upon a core design choice in neural combinatorial optimization: how should model complexity be distributed between the encoder and decoder? The prevalent heavy-encoder-light-decoder paradigm relies on a powerful encoder (such as the GNN you suggested) to generate information-dense static node representations. However, as discussed in our paper, such static representations struggle to capture the dynamic changes during the solving process when faced with large-scale problems, thereby limiting their generalization ability.
>
> In contrast, we adhere to a "Light-Encoder-Heavy-Decoder" (LEHD) design. The core idea is that a concise encoder is only responsible for basic feature extraction, while the true "intelligence" is embedded within a powerful heavy decoder. At each decoding step, the heavy decoder dynamically and iteratively re-evaluates and models the complex relationships among the remaining nodes based on the partially formed solution. This dynamic modeling approach has been proven to possess stronger generalization capabilities when handling large-scale problems.
>
> To further validate our design philosophy—that in the LEHD architecture, performance is primarily driven by the heavy decoder—we conducted an additional ablation study. We replaced our existing single-layer Transformer encoder with a more basic Multi-Layer Perceptron (MLP) encoder and re-evaluated the model's performance. The experimental results are shown in the table below:
>
> | Pro. | scale=100 | scale=100 | scale=500| scale=500| scale=1000| scale=1000  |
> | :--- | :----------------------- | :--------------------- | :----------------------- | :--------------------- | :----------------------- | :--------------------- |
> |      | MTLKD(MLP)               | MTLKD                  | MTLKD(MLP)               | MTLKD                  | MTLKD(MLP)               | MTLKD                  |
> | CVRP | 16.05 (3.37%)            | **16.04 (3.30%)**         | 64.63 (4.13%)            | **64.61 (4.09%)**          | **124.09 (3.80%)**           | 124.44 (4.09%)         |
> | VRPTW| 26.14 (7.37%)            | **26.13 (7.32%)**          | **99.04 (9.30%)**            | 99.05 (9.31%)          | 184.97 (11.12%)          | **184.92 (11.09%)**        |
> | VRPBL| 12.54 (4.33%)            | **12.54 (4.32%)**          | **45.59 (-4.88%)**           | 45.67 (-4.72%)         | 85.82 (-4.46%)           | **86.28 (-3.95%)**         |
> | VRPLTW| 26.38 (7.93%)            | **26.35 (7.81%)**          | 100.52 (9.42%)           | **100.40 (9.30%)**         | 192.61 (10.20%)          | **192.41 (10.08%)**        |
> | VRPBTW| 28.50 (12.16%)           | **28.49 (12.12%)**         | 104.27 (6.64%)           | **103.85 (6.21%)**         | 210.32 (8.03%)           | **209.82 (7.77%)**         |
>
> The results clearly demonstrate that even with the encoder simplified to an MLP, the overall model performance shows only minor changes. This strongly supports that, within our LEHD framework, the encoder's structure is not the performance bottleneck, and its role is relatively limited. Therefore, designing a GNN or other complex encoder architectures for our model is not only unnecessary but also inconsistent with our architectural philosophy, which aims to highlight the decoder's role.
>
> ---
>
> ### Weakness 2 & Questions 1 & 3: Clarity of Training Process and Figure Readability
> Thank you very much for raising these comments. We would like to make the following clarification to address your concerns.
>
> **W2:** The description of the training process for the student model is somewhat unclear, which may lead to confusion regarding its training effectiveness. Additionally, the text in the ablation study figures is too small and difficult to read.
>
> **Q1:** The training process of the student model, particularly how it distills and incorporates knowledge from the teacher models, is not clearly described. The current explanation may leave readers uncertain about the effectiveness and implementation details of the knowledge distillation mechanism. Try to make the training procedure of the student model more explicit in both the textual description and the framework figure.
>
> **Q3:** Some figures, especially in the ablation study section, contain text that is too small to read clearly. This affects the readability and interpretability of the experimental results. Improve the visual clarity of these figures would make the paper more accessible and reader-friendly.
>
> Thank you very much for pointing out the lack of clarity in the student model's training process description. We appreciate you highlighting this, as ensuring the transparency of our methodology is paramount. To help readers more accurately understand our multi-teacher knowledge distillation mechanism, we have elaborated on the training procedure as follows and have revised the text and figures accordingly in the final manuscript.
>
> In a training batch, we first construct instances that simultaneously include all N=6 types of visible VRP tasks. At any decoding step $t$, the student model $S$ outputs a unified probability distribution $\pi_{\theta^S}(a_t|s_t, \mathcal{G})$ based on the shared state $s_t$ of all instances. Concurrently, for each instance in the batch, we invoke its task-specific "expert" teacher model to perform a parallel forward pass under the exact same state $s_t$, thereby obtaining its guiding probability distribution. For example, a CVRP instance is guided by the CVRP teacher, and a VRPTW instance by the VRPTW teacher. Our objective is to align the student's behavior with the corresponding teacher's output by minimizing the KL divergence between them. The total loss is the sum of losses for all instances across all decoding steps within the batch. By optimizing this objective, the student model's single parameter set is driven to learn a unified policy that can imitate the respective experts based on different task types.
>
> ---
>
> Once again, we would like to express our heartfelt gratitude for the time and effort you've dedicated to reviewing our work. We sincerely hope that our response can address your concerns.

---

### Official Review · Reviewer_RweB · 2025-07-03

**Clarity:** 3
**Significance:** 2
**Originality:** 2
**Rating:** 4
**Confidence:** 4

**Summary:**

In their work, the authors attempt to mitigate the issue of poor generalization in multi-task learning for vehicle routing problems. They propose the use of knowledge distillation to train a heavy decoder model using the outputs of individual single-task models, each trained for a VRP variant, and then applying a loss on the difference in the outputted probability distributions. Additionally, to escape local optimas they propose the use of Random Reordering Re-Construction (R3C), which divides routes into smaller sub-routes and shuffles the internal order while respecting the problem constraints. They then study the generalization ability of the model on unseen VRP variants and unseen problem scales.

**Questions:**

1. What seperates your method from a direct application of [3] to MT Solvers?
2. Although not necessary for this paper, have you tried seeing if the method generalizes to the dynamic constraint problems as well?
3. Have you tried different sized subsets for R3C?

**Ethical Concerns:**

["NO or VERY MINOR ethics concerns only"]

**Final Justification:**

The exprimental section does seem more convincing now with the aligned ablation datasets and some baselines but I still have some reserves on the novelty of the approach.

**Limitations:**

yes

**Quality:**

2

**Strengths And Weaknesses:**

Strengths:

The knowledge distillation approach does seem to greatly improve generalization ability in most scenarios with good ablation experiments. Additionally, the R3C method is relatively novel and well founded in avoiding the local optimization minimas.

Weaknesses:

I do have a few concerns with the paper namely:
* The authors argue that this is a label free method, however this is not entirely correct since the teachers need to be trained on "labeled" data. It is more of a labeled training delegation.
* Some key baselines are omitted from the results. Namely [1, 2] are referenced in the paper but never compared against, despite being state-of-the-art multi-task VRP neural solvers and reporting better results (I could not find any in-text citation for [1] in the main paper either, only its reference in the bibliography).
* The novelty of knowledge distillation for VRP solvers is questionable since previous papers such as [3], which the author references, have previously shown the effectiveness of this approach in enhancing model generalization abilities. There was no actual discussion or comparison to it from a methodological perspective.
* There are some issues with formatting, but this is minor, specifically in table 1 for VRPL MT POMO, the number is unintelligible.

[1] Han Li, Fei Liu, Zhi Zheng, Yu Zhang, and Zhenkun Wang. Cada: Cross-problem routing297 solver with constraint-aware dual-attention. arXiv preprint arXiv:2412.00346, 2024.
[2] Federico Berto, Chuanbo Hua, Nayeli Gast Zepeda, André Hottung, Niels Wouda, Leon Lan,249 Junyoung Park, Kevin Tierney, and Jinkyoo Park. Routefinder: Towards foundation models for250 vehicle routing problems. arXiv preprint arXiv:2406.15007, 2024.
[3] Jieyi Bi, Yining Ma, Jiahai Wang, Zhiguang Cao, Jinbiao Chen, Yuan Sun, and Yeow Meng252 Chee. Learning generalizable models for vehicle routing problems via knowledge distillation.253 Advances in Neural Information Processing Systems, 35:31226–31238, 2022.

---

> ### Author Rebuttal · Authors · 2025-07-31
>
> Dear Reviewer RweB,
>
> Thank you very much for taking the time andeffort to review our work. We are delighted toknow you find that our paper is well written andrelatively easyto follow,that the idea of ageneric solver for several of the VRP variantsis well motivated, and that the use of knowledgedistillation for training the proposed modelmakes good sense. We address your concerns pointby-point as follows：
>
> ---
>
> ### W1: Clarifying "Label-Free" Methodology
>
> Thank you very much for raising this concern. We'd like to clarify that our teacher model is trained purely through RL, completely devoid of explicit optimal path labels. Similarly, the student model's distillation relies solely on the policy distribution from the teacher, eliminating the need for human or external labels throughout the entire framework. Therefore, our "label-free" training claim stands firm.
>
> ---
>
> ### W2: Additional baselines.
>
> Thank you very much for highlighting the omission of certain baselines. We've now conducted additional experiments, including comparisons with SOTA multi-task Vehicle Routing Problem (VRP) neural solvers, RF[2] and CaDA[1], on the same visible tasks and settings as our model. As **Table 1** shows, our model consistently demonstrates superior performance.
>
> Table 1: Performance Comparison of RF, CaDA, and MTL-KD on seen Tasks
> | Method | Problem Type | n=100 | n=500 | n=1k |
> | :----- | :----------- | :---- | :---- | :--- |
> | RF-Transformer | CVRP | 15.82(1.88%) | 67.71(9.08%) | 132.79(11.08%) |
> | RF-MTPOMO | CVRP | 15.87(2.20%) | 67.42(8.62%) | 132.82(11.11%) |
> | RF-MVMoE | CVRP | 15.84(2.01%) | 67.36(8.52%) | 134.85(12.80%) |
> | CaDA | CVRP | 15.84(2.00%) | 175.65(182.99%) | 542.56(353.87%) |
> | MTL-KD$\_{aug8}$ | CVRP | 15.85(2.08%) | 64.17(3.38%) | 123.76(3.53%) |
> | MTL-KD$\_{R3C}$ | CVRP | **15.76(1.48%)** | **63.63(2.51%)** | **122.06(2.10%)** |
> | RF-Transformer | VRPTW | 27.39(12.49%) | 108.70(19.97%) | 222.92(33.91%) |
> | RF-MTPOMO | VRPTW | 26.29(7.98%) | 102.77(13.42%) | 193.57(16.28%) |
> | RF-MVMoE | VRPTW | 26.29(7.98%) | 100.78(11.22%) | 187.87(12.86%) |
> | CaDA | VRPTW | 30.16(23.87%) | 300.11(231.21%) | 693.61(316.66%) |
> | MTL-KD$\_{aug8}$ | VRPTW | 25.72(5.62%) | 97.86(8.00%) | 183.39(10.17%) |
> | MTL-KD$\_{R3C}$ | VRPTW | **25.31(3.93%)** | **96.43(6.42%)** | **181.85(9.24%)** |
> | RF-Transformer | VRPL | 15.88(1.93%) | 68.59(7.93%) | 135.15(10.17%) |
> | RF-MTPOMO | VRPL | 15.93(2.23%) | 68.33(7.52%) | 134.53(9.66%) |
> | RF-MVMoE | VRPL | 15.90(2.06%) | 68.39(7.62%) | 138.60(12.98%) |
> | CaDA | VRPL | 15.89(2.00%) | 176.04(177.01%) | 536.71(337.49%) |
> | MTL-KD$\_{Single t}$ | VRPL | 15.92(2.12%) | 65.09(2.42%) | 125.73(2.48%) |
> | MTL-KD$\_{R3C}$ | VRPL | **15.82(1.50%)** | **64.52(1.53%)** | **124.59(1.56%)** |
> | RF-Transformer | OVRP | 10.11(4.11%) | 42.88(21.48%) | 84.40(27.68%) |
> | RF-MTPOMO | OVRP | 10.17(4.70%) | 41.60(17.86%) | 82.14(24.26%) |
> | RF-MVMoE | OVRP | 10.13(4.29%) | 41.18(16.65%) | 81.82(23.78%) |
> | CaDA | OVRP | 10.10(4.04%) | 187.72(431.78%) | 442.57(569.55%) |
> | MTL-KD$\_{Single t}$ | OVRP | 10.18(4.90%) | 38.34(8.63%) | 71.80(8.63%) |
> | MTL-KD$\_{R3C}$ | OVRP | **10.05(3.53%)** | **37.79(7.07%)** | **71.40(8.03%)** |
>
> ---
>
> ### W3 & Q1: Novelty of Knowledge Distillation and Comparison to AMDKD[3]
>
> Thank you very much for raising this concern. We want to clarify that our paper doesn't simply apply AMDKD to a multi-task scenario; instead, we introduce three key innovations:
>
> **Heterogeneous Teacher-Student Architecture:** Unlike AMDKD's homogeneous distillation, we propose **heterogeneous distillation** from a heavy encoder to a lightweight encoder. This allows us to leverage the low-cost training of the heavy encoder for generating soft probability distributions as supervision, while exploiting the heavy decoder's strong generalization for larger scales.
> **Label-Free Multi-Task Training Paradigm for Heavy Encoders:** We present a novel, **label-free multi-task training approach for the heavy decoder model**. This is achieved by allowing multiple lightweight decoder teacher models to undergo label-free reinforcement learning, with the heavy decoder student model directly distilling knowledge from them, opening a new avenue for training heavy encoders.
> **R3C Strategy for Enhanced Inference Performance:** We designed the **R3C strategy** specifically for MTL-KD. This strategy is adaptable to various VRP problems and significantly improves the model's inference performance.
>
> To provide a quantitative comparison, we also reproduced a multi-task extended version of AMDKD. As **Table 2** shows, our model demonstrates superior performance.
>
> Table 2: Performance Comparison of MTL-AMDKD and MTLKD Models
>
> | **Problem** | **MTLAMDKD (n=100)** | **MTLKD (n=100)** | **MTLAMDKD (n=500)** | **MTLKD (n=500)** | **MTLAMDKD (n=1000)** | **MTLKD (n=1000)** |
> | :---------- | :-------------------- | :----------------- | :-------------------- | :----------------- | :--------------------- | :------------------ |
> | CVRP | 16.10 (3.72%) | **16.04 (3.30%)** | 67.73 (9.11%) | **64.61 (4.09%)** | 134.22 (12.28%) | **124.44 (4.09%)** |
> | VRPTW | 26.13 (7.32%) | **26.13 (7.32%)** | 108.71 (19.98%) | **99.05 (9.31%)** | 224.93 (35.12%) | **184.92 (11.09%)** |
> | VRPB | **12.30 (2.74%)** | 12.38 (3.41%) | 49.90 (4.49%) | **45.99 (-3.70%)** | 101.76 (14.88%) | **86.99 (-1.78%)** |
> | VRPL | 16.16 (3.67%) | **16.12 (3.45%)** | 68.59 (7.94%) | **65.48 (3.03%)** | 136.22 (11.03%) | **126.66 (3.25%)** |
> | OVRP | **10.38 (6.95%)** | 10.46 (7.19%) | 42.23 (19.63%) | **38.89 (10.17%)** | 83.33 (26.07%) | **72.70 (9.98%)** |
> | OVRPTW | 15.25 (9.29%) | **15.08 (8.07%)** | 64.82 (34.61%) | **53.89 (11.91%)** | 137.34 (65.50%) | **93.96 (13.22%)** |
> | OVRPB | 9.72 (16.15%) | **9.27 (10.81%)** | 33.97 (13.32%) | **30.45 (1.58%)** | 64.87 (18.23%) | **56.92 (3.73%)** |
> | OVRPL | **10.34 (6.95%)** | 10.44 (7.95%) | 41.59 (19.87%) | **38.43 (10.76%)** | 82.84 (26.68%) | **72.78 (11.31%)** |
> | VRPBL | **12.37 (2.91%)** | 12.54 (4.32%) | 49.70 (3.69%) | **45.67 (-4.72%)** | 100.72 (12.14%) | **86.28 (-3.95%)** |
> | VRPBTW | **27.64 (8.79%)** | 28.49 (12.12%) | 111.25 (13.78%) | **103.85 (6.21%)** | 245.08 (25.89%) | **209.82 (7.77%)** |
> | VRPLTW | **26.23 (7.32%)** | 26.35 (7.81%) | 109.92 (19.65%) | **100.40 (9.30%)** | 230.50 (31.87%) | **192.41 (10.08%)** |
> | OVRPBL | 9.70 (16.22%) | **9.41 (12.75%)** | 33.94 (14.65%) | **31.42 (6.14%)** | 64.61 (18.97%) | **59.25 (9.11%)** |
> | OVRPBTW | **16.40 (14.02%)** | 16.74 (16.39%) | 65.54 (26.32%) | **56.02 (7.98%)** | 138.67 (52.61%) | **99.20 (9.18%)** |
> | OVRPLTW | 15.30 (9.25%) | **15.26 (8.95%)** | 64.61 (34.68%) | **53.56 (11.65%)** | 137.44 (64.24%) | **95.33 (13.92%)** |
> | VRPBLTW | 27.97 (10.36%) | **29.13 (14.96%)** | 116.66 (13.08%) | **110.33 (6.94%)** | 239.00 (26.25%) | **205.52 (8.56%)** |
> | OVRPBLTW | **16.24 (13.98%)** | 16.95 (18.93%) | 65.94 (25.91%) | **57.15 (9.13%)** | 138.73 (50.53%) | **101.64 (10.29%)** |
> ---
>
> ### W4: Formatting Issues
>
> Thank you for your comments; we will fix all formatting issues.
>
> ---
>
> ### Q2: Generalization to Dynamic Constraint Problems
>
> Thank you very much for this valuable suggestion. While our heavy decoder's dynamic 're-encoding' concept naturally aligns with dynamic VRPs, adapting our current static model poses significant challenges. It's not a simple fine-tuning; it requires fundamental architectural changes for dynamic input and, crucially, building a new simulation environment for retraining from scratch. This is a substantial undertaking we can't complete now, but it's a valuable direction for future research.
>
> ---
>
> ### Q3: Different Sized Subsets for R3C
>
> Thank you very much for this insightful question. To understand how different subset sizes affect our R3C strategy, we ran an extra experiment. We systematically compared fixed sampling lengths (10, 20, 30, 40, 50) against our paper's random size strategy on 100-node instances for four core VRP variants. As Table 3 details, all results reflect performance after 50 iterations.
>
> The results clearly show that a random subset size strategy consistently outperforms fixed sizes across all tested problems, delivering the best performance. We believe this is because random sizes generate a more diverse range of subproblems. Fixed sizes can lead the model to get stuck in local optima by repeatedly processing similar structures, while random sizes help it escape local optima and find better solutions.
>
> Table 3: Results after 50 iterations of R3C with different sampling lengths on scale 100
>
> | **Problem** | **subset=10** | **subset=20** | **subset=30** | **subset=40** | **subset=50** | **random** |
> | :---------- | :------------ | :------------ | :------------ | :------------ | :------------ | :--------- |
> | CVRP | 16.01 | 15.96 | 15.94 | 15.91 | 15.91 | **15.88** |
> | OVRP | 10.34 | 10.25 | 10.21 | 10.17 | 10.20 | **10.17** |
> | VRPL | 16.09 | 16.03 | 16.00 | 15.97 | 15.98 | **15.90** |
> | VRPTW | 26.02 | 25.86 | 25.77 | 25.65 | 25.69 | **25.55**|
>
>
> [1] Han Li, Fei Liu, Zhi Zheng, Yu Zhang, and Zhenkun Wang. Cada: Cross-problem routing solver with constraint-aware dual-attention. ICML, 2025.
>
> [2] Federico Berto, Chuanbo Hua, Nayeli Gast Zepeda, André Hottung, Niels Wouda, Leon Lan, Junyoung Park, Kevin Tierney, and Jinkyoo Park. Routefinder: Towards foundation models for vehicle routing problems. ArXiv, 2024.
>
> [3] Jieyi Bi, Yining Ma, Jiahai Wang, Zhiguang Cao, Jinbiao Chen, Yuan Sun, and Yeow Meng Chee. Learning generalizable models for vehicle routing problems via knowledge distillation. NeurIPS, 2022.
>
> ---
>
> Once again, we would like to express our heartfelt gratitude for the time and effort you've dedicated to reviewing our work. We sincerely hope that our response can address your concerns.

---

> > ### Comment · Reviewer_RweB · 2025-08-02
> >
> > Thank you very much for the time and effort you've put into the rebuttal. The experimental section seems more reasonable now and I will adjust my score accordingly.

---

> > > ### Author Response · Authors · 2025-08-05
> > > **Thank you**
> > >
> > > Thank you very much for your effort in reviewing our paper and engaging with us in the discussion. We are glad to know you feel the experimental section more reasonable and you will adjust your score accordingly.

---

> > > > ### Comment · Reviewer_6wtN · 2025-08-09
> > > >
> > > > I appreciate the efforts from the authors but I would like to maintain my previous rating because the large performance gap compared with exact methods.

---

### Official Review · Reviewer_6wtN · 2025-07-04

**Clarity:** 3
**Significance:** 3
**Originality:** 2
**Rating:** 3
**Confidence:** 4

**Summary:**

This paper introduces a multi-task learning method called MTL-KD that enables efficient training of heavy decoder models with strong generalization ability for solving diverse Vehicle Routing Problem (VRP) variants. The main works include 1)  label-free training of heavy decoder models through knowledge distillation from multiple RL-based single-task teacher models; 2) a Random Reordering Reconstruction (R3C) strategy to further enhance the performance of the multi-task model; Experimental results show superior performance of the MTL-KD model on both seen and unseen VRP tasks, as well as real-world datasets, exhibiting good scale generalization ability and outperforming existing multi-task VRP models.

**Questions:**

What are the clear benefits of neural VRP solvers in comparison with traditional VRP methods from OR community (e.g. branch-and-price-and-cut methods) and metaheuristics (e.g. genetic algorithm, VLNS, etc.)?

Why is the Random Reordering Reconstruction (R3C) strategy used? Does this imply weak performance by the proposed neural solver?

**Ethical Concerns:**

["NO or VERY MINOR ethics concerns only"]

**Final Justification:**

I would like to maintain the previous rating on the balance of the positive changes and responses from the authors but remain concerned with the large performance gaps of the proposed method when compared with the SOTA exact method.

**Limitations:**

The motivation of using neural solver for VRP could be better explained and the use of iterative procedure of R3C needs further justification  and the fairness of comparisons with other methods is questionable as many other methods do not use local search style improvement procedure.

**Quality:**

3

**Strengths And Weaknesses:**

Strength:
The paper is well written and relatively easy to follow. The idea of a generic solver for several of VRP variants is well motivated. The use of knowledge distillation for training the proposed model makes good sense.

Weakness:
1) the introduction fails to explain why the neural solvers are preferred compared to the traditional VRP solvers. As was shown in the paper and the results by the current SOTA neural VRP solvers, the gap of these results to the best heuristic/OR methods is still quite large (sometime as high as 5% or higher). Even for large scale instances, the metaheuristics have proven to be very effective. What are the clear benefits of neural VRP solvers? It is not clearly stated in the paper.

2) the proposed method uses a Random Reordering Reconstruction (R3C) strategy to further improve the performance. However, this would make the proposed method similar to an iterative improvement method widely used in metaheuristics. Therefore, a comparison with the state of the art metaheuristic for VRP should be necessary.

---

> ### Author Rebuttal · Authors · 2025-07-31
>
> Dear Reviewer 6wtN,
>
> Thank you for your time and effort in reviewing our work. We're glad you found our paper well-written, our experiments convincing in addressing existing NCO limitations, and our analysis insightful. We also appreciate your recognition of the proposed approach's significant improvements.
>
> We address your concerns point-by-point as follows.
>
> ---
>
> ### W1 & Q1: Why are neural solvers preferred compared to traditional VRP solvers? What are their clear benefits?
>
> Thank you very much for raising this concern. While classical heuristics still hold a slight edge in solution quality, their performance is highly dependent on expert-designed operators and meticulous parameter tuning, leading to very high deployment costs. Neural solvers, on the other hand, replace manual heuristic construction with **end-to-end learning**, requiring only data for training, which significantly **reduces the reliance on domain knowledge**.
>
> Furthermore, traditional methods experience a sharp increase in online solution time as the problem scale grows. In contrast, neural solvers, after a one-time offline training, can provide feasible solutions in a very short amount of time. This makes them particularly valuable for **real-time scheduling** and other scenarios requiring rapid responses, where their practical utility is significantly higher.
>
> ---
>
> ### W2: The proposed method uses R3C, making it similar to iterative improvement. A comparison with state-of-the-art metaheuristics is necessary.
>
> Thank you very much for raising this concern. As demonstrated in the experimental section of our paper, we have already compared our proposed method with metaheuristic solvers such as **PyVRP** and **OR-Tools**, which are widely regarded as strong baselines. Our method even **outperformed OR-Tools in some scenarios**.
>
> In our framework, the R3C strategy serves merely as an **optional enhancement module**. Even without enabling this strategy, our model itself surpasses other multi-task neural solvers. Therefore, our research focus remains on **elevating the inherent capabilities of Neural Combinatorial Optimization (NCO) methods**, rather than degenerating into a traditional metaheuristic iterative improvement framework.
>
> ---
>
> ### Q2: Why is the Random Reordering Reconstruction (R3C) strategy used? Does this imply weak performance by the proposed neural solver?
>
> Thank you very much for raising this concern. We would like to clarify that the use of the R3C strategy does **not imply insufficient performance** of our proposed neural solver. Instead, it is an **optional post-processing method** designed to further enhance performance, essentially "trading time for accuracy."
>
> Our model already significantly outperforms existing neural solvers in greedy decoding mode (as shown in Table 1), demonstrating its **strong inherent performance**. The introduction of R3C is intended to further unlock the potential of neural solvers and provide users with a **simple, general, and scalable way to improve solution accuracy**. Unlike methods such as LEHD-RRC, which are only applicable to TSP/CVRP, R3C can be seamlessly applied to **various VRP variants**, offering greater versatility.
>
> Table 1: Average performance of multi-task models on visible tasks. **Single t**: Results obtained via single trajectory inference. **aug8**: Results obtained using data augmentation (8 augmentations).
>
> | **Method** | **100 (Single t)** | **100 (aug8)** | **500 (Single t)** | **500 (aug8)** | **1000 (Single t)** | **1000 (aug8)** |
> | :-------- | :------------- | :----------- | :------------- | :----------- | :-------------- | :----------- |
> | MTPOMO | 7.5217% | 3.3900% | 23.6533% | 17.3883% | 40.3667% | 27.6867% |
> | MVMOE | 7.9433% | **3.0600%** | 41.0917% | 32.1633% | 91.9567% | 74.3500% |
> | MTLKD | **5.4569%** | 3.5920% | **5.8028%** | **4.5730%** | **6.6413%** | **5.6071%** |
>
> ---
>
> ### Limitations: Motivation of using neural solver, justification of R3C, and fairness of comparisons.
>
> Thank you very much for raising this concern. We have provided a clearer explanation for the motivation behind using VRP neural solvers in our response to **Q1/W1**. The justification for using R3C has been elaborated in our response to **Q2/W2**.
>
> Additionally, to ensure fairness in comparisons, we will supplement the main text with inference results obtained using aug8 with greedy decoding, consistent with practices in other related papers, as shown in Table 2. Compared to the other methods we evaluated, our model demonstrates superior performance in most scenarios.
>
> Table 2: MTLKD Performance with aug8 Augmentation under Greedy Decoding
> | **Problem Type** | **Scale 100** | **Scale 500** | **Scale 1000** | **Problem Type** | **Scale 100** | **Scale 500** | **Scale 1000** |
> | :----------- | :--------------- | :--------------- | :--------------- | :----------- | :--------------- | :--------------- | :--------------- |
> | CVRP | 15.850 (2.077%) | 64.169 (3.379%) | 123.759 (3.526%) | VRPTW | 25.718 (5.619%) | 97.859 (8.000%) | 183.389 (10.166%)|
> | OVRP | 10.184 (4.900%) | 38.343 (8.629%) | 71.803 (8.633%) | VRPB | 12.101 (1.104%) | 45.396 (-4.947%) | 85.814 (-3.118%) |
> | VRPL | 15.915 (2.118%) | 65.093 (2.425%) | 125.729 (2.484%) | OVRPTW | 14.753 (5.734%) | 52.942 (9.952%) | 92.902 (11.951%) |
> | OVRPL | 10.200 (5.494%) | 37.809 (8.959%) | 71.556 (9.433%) | VRPLTW | 25.910 (6.024%) | 99.400 (8.203%) | 190.778 (9.149%) |
> | OVRPLTW | 14.907 (6.468%) | 52.772 (10.015%) | 94.115 (12.464%) | OVRPB | 9.270 (10.810%) | 29.662 (-1.063%) | 54.805 (-0.115%) |
> | VRPBL | 12.178 (1.345%) | 45.067 (-5.974%) | 84.687 (-5.718%) | VRPBTW | 27.602 (8.637%) | 101.916 (4.237%) | 238.529 (22.520%)|
> | OVRPBL | 9.037 (8.249%) | 30.491 (2.997%) | 57.506 (5.896%) | OVRPBTW | 16.135 (12.173%) | 54.724 (5.476%) | 97.596 (7.410%) |
> | VRPBLTW | 28.093 (10.857%) | 108.349 (5.025%) | 202.361 (6.892%) | OVRPBLTW | 16.203 (13.711%) | 55.639 (6.250%) | 99.444 (7.902%) |
>
> ---
>
> Once again, we would like to express our heartfelt gratitude for the time and effort you've dedicated to reviewing our work. We sincerely hope that our response can address your concerns.

---

> > ### Comment · Reviewer_6wtN · 2025-08-05
> >
> > I cannot agree with the authors that "... classical heuristics still hold a slight edge in solution quality..." because the gap can be as high as 5%.  It is also not fair to accuse "parameter tuning and expertise design" in heuristic methods while the same issues apply to the proposed neural solvers. Additionally, the (meta)heuristics would perform  consistently across different data distributions and sizes while most existing neural solvers suffer from generalization issues. I hope the community stop publishing this type of paper as it does not advance the research because of wrong focus.

---

> > > ### Author Response · Authors · 2025-08-05
> > > **Response Regarding the Research Value of NCO and the Positioning of Our Work**
> > >
> > > Dear Reviewer 6wtN,
> > >
> > > Thank you very much for your valuable feedback in the discussion. We deeply respect your expertise and have carefully reflected on each of your points. We hope the following explanation can more clearly communicate the motivation, positioning, and contributions of our research.
> > >
> > > **(1) The Performance Gap and Research Focus of NCO vs. Traditional Heuristics:**
> > >
> > > We fully agree with your point that the current state-of-the-art traditional heuristics (e.g., HGS, LKH3) indeed outperform most NCO methods in solution quality on many classic VRP tasks. Describing this gap as "slight" was not sufficiently rigorous. We sincerely apologize for this imprecise statement and will correct it in the final version.
> > >
> > > We wish to clarify that our research focus is not to completely surpass a mature field developed over decades in the short term, but rather to explore a different and highly promising technological path:
> > >
> > > - **Difference in Developmental Stages:** Traditional heuristics have undergone decades of in-depth research, leading to a rich set of highly effective strategies tailored to specific problem properties (e.g., Variable Neighborhood Search). In contrast, NCO is an emerging field that gained widespread attention with Pointer Networks [1] in 2015 and was combined with the powerful Transformer framework only in 2018 [2]. It may be unrealistic to expect a newborn field to comprehensively outperform a mature one in such a short period.
> > >
> > > - **The Rapid Catch-up of the NCO Field:** Despite the gap, the NCO community is striving to catch up and has already demonstrated significant potential in specific scenarios. For instance, on the CVRP problem you highlighted, single-task NCO methods have already achieved breakthroughs. A recent study [3] has also surpassed HGS on problems like CVRP and VRPTW. This indicates that the potential of NCO methods is gradually being realized.
> > >
> > > - **Positioning of Our Work: Exploring Unified Multi-Task Solving:** Our work focuses on a more challenging and emerging subfield—**multi-task NCO**. Pioneering work in this direction was only formally published in 2024 [4], making the field itself only about two years old. Our objective is to develop a **unified** model capable of solving multiple VRP variants simultaneously, thereby minimizing the cost of repetitive manual design and algorithm development for each variant. As noted in our paper, the real world presents a vast number of VRP variants (our study alone covers 16), and this diversity highlights the significant value of developing a unified framework that can generalize across different problem structures.
> > >
> > > **(2) "Parameter Tuning" and "Model Generalization":**
> > >
> > > We completely agree that NCO models also require "parameter tuning and expertise design". Expertise is needed for network architecture design, hyperparameter selection, and training strategies. The goal of NCO is to transform this expertise into the design of a **general-purpose** machine learning framework. We believe that a well-designed NCO framework has the potential to automatically capture the intrinsic structures of different problems through learning on multi-task data. Consequently, when faced with new tasks or distributions, it would require as little as possible "from-scratch" design effort.
> > >
> > > While the generalization of current NCO models remains a significant challenge, it is also a primary motivation for our work. Our research is precisely aimed at enhancing the model's generalization capabilities in a multi-task environment.
> > >
> > > **(3) Our Contributions and Vision:**
> > >
> > > Although we acknowledge that our method still has a gap with state-of-the-art single-task solvers like HGS, our work represents solid progress within the NCO field. In our experiments, our unified model's performance on several complex, combined-constraint instances (such as OVRPB500 and OVRPBL500) already surpasses that of OR-Tools, a widely used classic solver. This demonstrates the effectiveness of our unified model framework and provides a valuable reference for the NCO community.
> > >
> > > Our vision, shared by many researchers in the NCO community, is to ultimately achieve a unified, learning-based model that minimizes manual design, effectively solves multiple problems, and whose performance can approach or even surpass that of traditional heuristics. We believe our work is a meaningful step toward this ambitious goal, capable of advancing the field and inspiring future research.
> > >
> > > Thank you again for your valuable time and insightful feedback. We will carefully revise the related phrasing in our final manuscript based on your comments to position our work more clearly and objectively.
> > >
> > > [1] Pointer networks. NeurIPS 2015.
> > >
> > > [2] Attention, Learn to Solve Routing Problems! ICLR 2019.
> > >
> > > [3] Neural Deconstruction Search for Vehicle Routing Problems. Arxiv 2025.
> > >
> > > [4] Multi-Task Learning for Routing Problem with Cross-Problem Zero-Shot Generalization. KDD 2024.

---

> ### Comment · Reviewer_RweB · 2025-08-06
>
> Yeah on this issue, I also wouldn't disregard the deep learning approaches because they hold promise in advancing heuristic methods such as in [1]. And while pure neural methods still fall behind in terms of optimality gaps, they are better suited for time sensitive scenarios.
>
>  Overall, the insights gained from these approaches can be employed in hybrid heuristic architectures. And in general this is a valid direction where even exploratory research that shows what doesnt work is beneficial in itself (so long as the approach is well justified and aligns with state of the art methods in other applications etc...). Although papers that show improved performances are obviously "more relevant" in terms of publication importance.
>
>
> [1] Xin, Liang, et al. "Neurolkh: Combining deep learning model with lin-kernighan-helsgaun heuristic for solving the traveling salesman problem." Advances in Neural Information Processing Systems 34 (2021): 7472-7483.

---

### Decision · Program_Chairs · 2025-09-17

**Decision:**

Accept (poster)

**Comment:**

This paper introduces a multi-task learning framework which uses knowledge distillation to train a single neural solver that can generalize to multiple variants of the Vehicle Routing Problem (VRP).

The reviewers agreed that the paper is well-written, and the method is technically sound with strong empirical results. The initial concerns were around the lack of some baselines, the clarity of the training process, and a more fundamental critique on the value of neural solvers given their performance gap with traditional exact methods. The authors addressed nearly all concerns with additional results.
Although there was a criticism about the ML-based approach given the existence of exact algorithms, I believe that Neural Combinatorial Optimization (NCO) is generally a well-established research field. Thus, it is not a sufficient reason to reject a paper.

Given the paper's technical contributions and comprehensive empirical validation, I recommend acceptance. I encourage the authors to incorporate a more detailed motivation for NCO in their camera-ready version to address the broader context raised by Reviewer 6wtN.